# DeepFly3D, a deep learning-based approach for 3D limb and appendage tracking in tethered, adult *Drosophila*

Semih Günel[1,2]*, Helge Rhodin[1,3], Daniel Morales[2], João Campagnolo[2], Pavan Ramdya[2†]*, Pascal Fua[1†]

[1]Computer Vision Laboratory, School of Computer and Communication Sciences, EPFL, Lausanne, Switzerland; [2]Neuroengineering Laboratory, Brain Mind Institute & Interfaculty Institute of Bioengineering, School of Life Sciences, EPFL, Lausanne, Switzerland; [3]Department of Computer Science, UBC, Vancouver, Canada

**Abstract** Studying how neural circuits orchestrate limbed behaviors requires the precise measurement of the positions of each appendage in three-dimensional (3D) space. Deep neural networks can estimate two-dimensional (2D) pose in freely behaving and tethered animals. However, the unique challenges associated with transforming these 2D measurements into reliable and precise 3D poses have not been addressed for small animals including the fly, *Drosophila melanogaster*. Here, we present DeepFly3D, a software that infers the 3D pose of tethered, adult *Drosophila* using multiple camera images. DeepFly3D does not require manual calibration, uses pictorial structures to automatically detect and correct pose estimation errors, and uses active learning to iteratively improve performance. We demonstrate more accurate unsupervised behavioral embedding using 3D joint angles rather than commonly used 2D pose data. Thus, DeepFly3D enables the automated acquisition of *Drosophila* behavioral measurements at an unprecedented level of detail for a variety of biological applications.
DOI: https://doi.org/10.7554/eLife.48571.001

*For correspondence:
semih.gunel@epfl.ch (SG);
pavan.ramdya@epfl.ch (PR)

†These authors contributed
equally to this work

Competing interests: The
authors declare that no
competing interests exist.

Reviewing editor: Timothy
O'Leary, University of
Cambridge, United Kingdom

## Introduction

The precise quantification of movements is critical for understanding how neurons, biomechanics, and the environment influence and give rise to animal behaviors. For organisms with skeletons and exoskeletons, these measurements are naturally made with reference to 3D joint and appendage locations. Paired with modern approaches to simultaneously record the activity of neural populations in tethered, behaving animals (*Dombeck et al., 2007*; *Seelig et al., 2010*; *Chen et al., 2018*), 3D joint and appendage tracking promises to accelerate the discovery of neural control principles, particularly in the genetically tractable and numerically simple nervous system of the fly, *Drosophila melanogaster*.

However, algorithms for reliably estimating 3D pose in such small *Drosophila*-sized animals have not yet been developed. Instead, multiple alternative approaches have been taken. For example, one can affix and use small markers—reflective, colored, or fluorescent particles—to identify and reconstruct keypoints from video data (*Bender et al., 2010*; *Kain et al., 1910*; *Todd et al., 2017*). Although this approach works well on humans (*Moeslund and Granum, 2000*), in smaller, *Drosophila*-sized animals markers likely hamper movements and are difficult to mount on sub-millimeter scale limbs. Most importantly, measurements of one or even two markers for each leg (*Todd et al., 2017*) cannot fully describe 3D limb kinematics. Another strategy has been to use computer vision techniques that operate without markers. However, these measurements have been restricted to 2D pose in freely behaving flies. Before the advent of deep learning, this was accomplished by matching the

contours of animals seen against uniform backgrounds (*Isakov et al., 2016*), measuring limb tip positions using complex TIRF-based imaging approaches (*Mendes et al., 2013*), or measuring limb segments using active contours (*Uhlmann et al., 2017*). In addition to being limited to 2D rather than 3D pose, these methods are complex, time-consuming, and error-prone in the face of long data sequences, cluttered backgrounds, fast motion, and occlusions that naturally occur when animals are observed from a single 2D perspective.

As a result, in recent years the computer vision community has largely forsaken these techniques in favor of deep learning-based methods. Consequently, the efficacy of monocular 3D human pose estimation algorithms has greatly improved. This is especially true when capturing human movements for which there is enough annotated data to train deep networks effectively. Walking and upright poses are prime examples of this, and state-of-the-art algorithms (*Pavlakos et al., 2017a*; *Tome et al., 2017*; *Popa et al., 2017*; *Moreno-noguer, 2017*; *Martinez et al., 2017*; *Mehta et al., 2017*; *Rogez et al., 2017*; *Pavlakos et al., 2017b*; *Zhou et al., 2017*; *Tekin et al., 2017*; *Sun et al., 2017*) now deliver impressive real-time results in uncontrolled environments. Increased robustness to occlusions can be obtained by using multi-camera setups (*Elhayek et al., 2015*; *Rhodin et al., 2016*; *Simon et al., 2017*; *Pavlakos et al., 2017b*) and triangulating the 2D detections. This improves accuracy while making it possible to eliminate false detections.

These advances in 2D pose estimation have also recently been used to measure behavior in laboratory animals. For example, DeepLabCut provides a user-friendly interface to DeeperCut, a state-of-the-art human pose estimation network (*Mathis et al., 2018*), and LEAP (*Pereira et al., 2019*) can successfully track limb and appendage landmarks using a shallower network. Still, 2D pose provides an incomplete representation of animal behavior: important information can be lost due to occlusions, and movement quantification is heavily influenced by perspective.

Approaches used to translate human 2D to 3D pose have also been applied to larger animals, like lab mice and cheetahs (*Nath et al., 2019*), but require the use of calibration boards. These techniques cannot be easily transferred for the study of small animals like *Drosophila*: adult flies are approximately 2.5 mm long and precisely registering multiple camera viewpoints using traditional approaches would require the fabrication of a prohibitively small checkerboard pattern, along with the tedious labor of using a small, external calibration pattern. Moreover, flies have many appendages and joints, are translucent, and in most laboratory experiments are only illuminated using infrared light (to avoid visual stimulation)—precluding the use of color information.

To overcome these challenges, we introduce DeepFly3D, a deep learning-based software pipeline that achieves comprehensive, rapid, and reliable 3D pose estimation in tethered, behaving adult *Drosophila* (*Figure 1*, *Figure 1—video 1*). DeepFly3D is applied to synchronized videos acquired from multiple cameras. It first uses a state-of-the-art deep network (*Newell et al., 2016*) and then enforces consistency across views. This makes it possible to eliminate spurious detections, achieve high 3D accuracy, and use 3D pose errors to further fine-tune the deep network to achieve even better accuracy. To register the cameras, DeepFly3D uses a novel calibration mechanism in which the fly itself is the calibration target. During the calibration process, we also employ sparse bundle adjustment methods, as previously used for human pose estimation (*Takahashi et al., 2018*; *Triggs et al., 2000*; *Puwein et al., 2014*). Thus, the user does not need to manufacture a prohibitively small calibration pattern, or repeat cumbersome calibration protocols. We explain how users can modify the codebase to extend DeepFly3D for 3D pose estimation in other animals (see Materials and methods). Finally, we demonstrate that unsupervised behavioral embedding of 3D joint angle data is robust against problematic artifacts present in embeddings of 2D pose data. In short, DeepFly3D delivers 3D pose estimates reliably, accurately, and with minimal manual intervention while also providing a critical tool for automated behavioral data analysis.

## Results

### DeepFly3D

The input to DeepFly3D is video data from seven cameras. These images are used to identify the 3D positions of 38 landmarks per animal: (i) five on each limb – the thorax-coxa, coxa-femur, femur-tibia, and tibia-tarsus joints as well as the pretarsus, (ii) six on the abdomen - three on each side, and (iii)

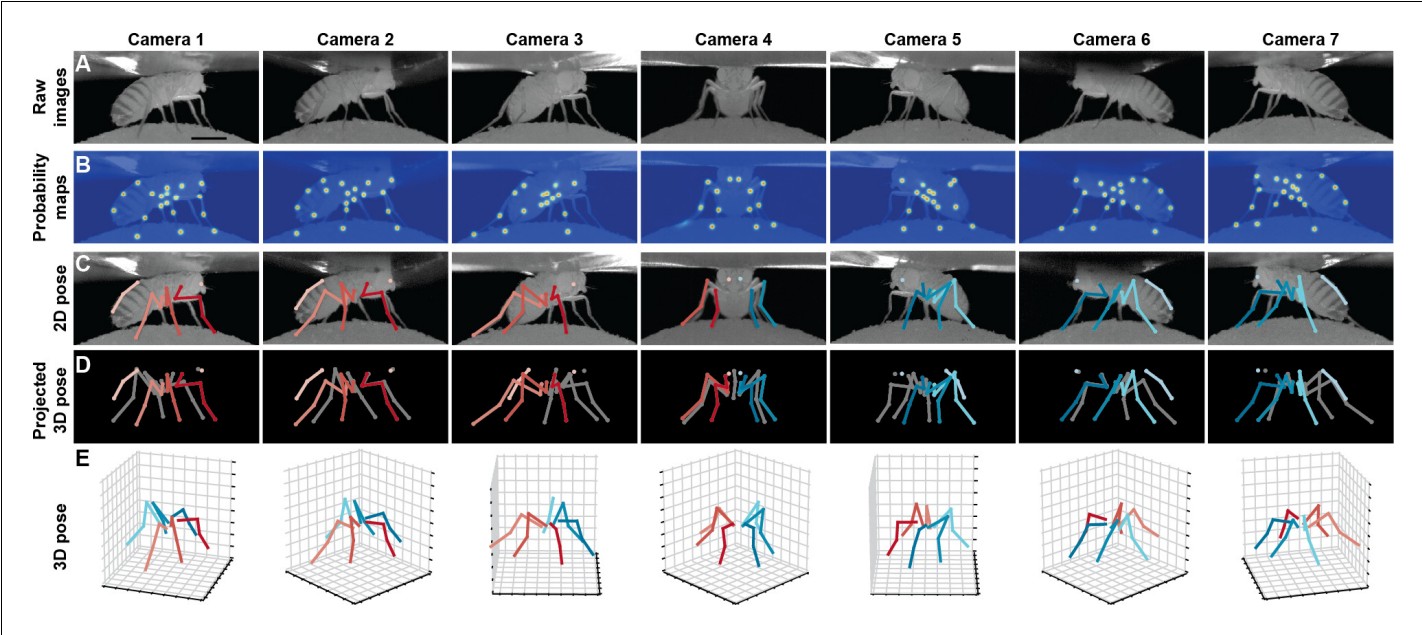

**Figure 1.** Deriving 3D pose from multiple camera views. (**A**) Raw image inputs to the Stacked Hourglass deep network. (**B**) Probability maps output from the trained deep network. For visualization purposes, multiple probability maps have been overlaid for each camera view. (**C**) 2D pose estimates from the Stacked Hourglass deep network after applying pictorial structures and multi-view algorithms. (**D**) 3D pose derived from combining multiple camera views. For visualization purposes, 3D pose has been projected onto the original 2D camera perspectives. (**E**) 3D pose rendered in 3D coordinates. Immobile thorax-coxa joints and antennal joints have been removed for clarity.

DOI: https://doi.org/10.7554/eLife.48571.002

The following video is available for figure 1:

**Figure 1—video 1.** Deriving 3D pose from multiple camera views during backward walking in an optogenetically stimulated MDN>CsChrimson fly.

DOI: https://doi.org/10.7554/eLife.48571.003

one on each antenna - for measuring head rotations. Our software incorporates the following innovations designed to ensure automated, high-fidelity, and reliable 3D pose estimation.

## Calibration without an external calibration pattern

Estimating 3D pose from multiple images requires calibrating the cameras to achieve a level of accuracy commensurate with the target size—a difficult challenge when measuring leg movements for an animal as small as *Drosophila*. Therefore, instead of using a typical external calibration grid, DeepFly3D uses the fly itself as a calibration target. It detects arbitrary points on the fly's body and relies on bundle-adjustment (*Chavdarova et al., 2018*) to simultaneously assign 3D locations to these points and to estimate the positions and orientations of each camera. To increase robustness, it enforces geometric constraints that apply to tethered flies with respect to limb segment lengths and ranges of motion.

## Geometrically consistent reconstructions

Starting with a state-of-the-art deep network for 2D keypoint detection in individual images (*Newell et al., 2016*), DeepFly3D enforces geometric consistency constraints across multiple synchronized camera views. When triangulating 2D detections to produce 3D joint locations, it relies on pictorial structures and belief propagation message passing (*Felzenszwalb and Huttenlocher, 2005*) to detect and further correct erroneous pose estimates.

## Self-supervision and active learning

DeepFly3D also uses multiple view geometry as a basis for active learning. Thanks to the redundancy inherent in obtaining multiple views of the same animal, we can detect erroneous 2D predictions for correction that would most efficiently train the 2D pose deep network. This approach greatly

reduces the need for time-consuming manual labeling (*Simon et al., 2017*). We also use pictorial structure corrections to fine-tune the 2D pose deep network. Self-supervision constitutes 85% of our training data.

## 2D pose performance and improvement using pictorial structures

We validated our approach using a challenging dataset of 2,063 image frames manually annotated using the DeepFly3D annotation tool and sampled uniformly from each camera. Images for testing and training were 480 × 960 pixels. The test dataset included challenging frames and occasional motion blur to increase the difficulty of pose estimation. For training, we used a final training dataset of 37,000 frames, an overwhelming majority of which were first automatically corrected using pictorial structures. On test data, we achieved a Root Mean Square Error (RMSE) of 13.9 pixels. Compared with a ground truth RMSE of 12.4 pixels – via manual annotation of 210 images by a new human expert – our Network Annotation/Manual Annotation ratio of 1.12 (13.9 pixels / 12.4 pixels) is similar to the ratio of another state-of-the-art network (*Mathis et al., 2018*): 1.07 (2.88 pixels / 2.69 pixels). Setting a 50 pixel threshold (approximately one third the length of the femur) for PCK (percentage of correct keypoints) computation, we observed a 98.2% general accuracy before applying pictorial structures. Notably, if we reduced our threshold to 30 or 20 pixels, we still achieved 95% or 89% accuracy, respectively (*Figure 2A*).

To test the performance of our network in a low data regime, we trained a two-stacked network using ground-truth annotations data from seven cameras (*Figure 2B*). We compared the results to an asymptotic prediction error (i.e. the error observed when the network is trained using the full

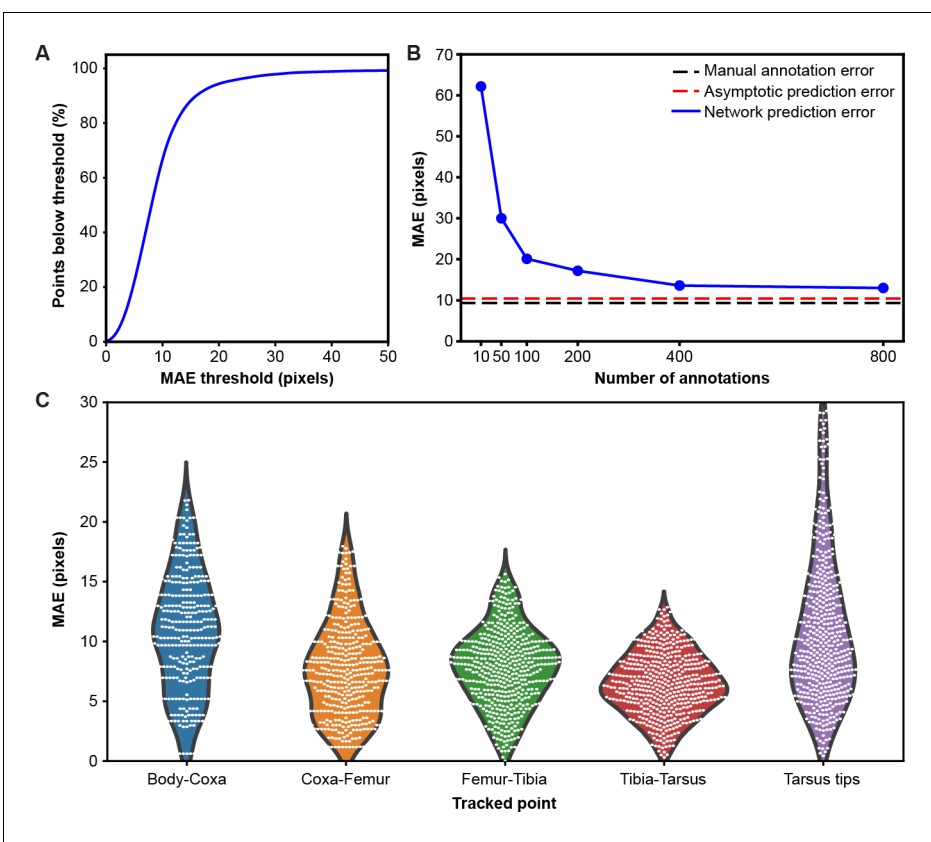

**Figure 2.** Mean absolute error distribution. (**A**) PCK (percentage of keypoints) accuracy as a function of mean absolute error (MAE) threshold. (**B**) Evaluating network prediction error in a low data regime. The Stacked Hourglass network (blue circles) shows near asymptotic prediction error (red dashed line), even when trained with only 400 annotated images. After 800 annotations, there are minimal improvements to the MAE. (**C**) MAE for different limb landmarks. Violin plots are overlaid with raw data points (white circles).
DOI: https://doi.org/10.7554/eLife.48571.004

dataset of 40,000 annotated images) and to the variability observed in human annotations of 210 randomly selected images. We measured an asymptotic MAE (mean absolute error) of 10.5 pixels and a human variability MAE of 9.2 pixels. With 800 annotations, our network achieved a similar accuracy to manual annotation and was near the asymptotic prediction error. Further annotation yielded diminishing returns.

Although our network achieves high accuracy, the error is not isotropic (*Figure 2C*). The tarsus tips (i.e. pretarsus) exhibited larger error than the other joints, perhaps due to occlusions from the spherical treadmill, and higher positional variance. Increased error observed for body-coxa joints might be due to the difficulty of annotating these landmarks from certain camera views.

To correct the residual errors, we applied pictorial structures. This strategy fixed 59% of the remaining erroneous predictions, increasing the final accuracy to 99.2%, from 98.2%. These improvements are illustrated in *Figure 3*. Pictorial structure failures were often due to pose ambiguities resulting from heavy motion blur. These remaining errors were automatically detected with multi-view redundancy using *Equation 6*, and earmarked for manual correction using the DeepFly3D GUI.

## 3D pose permits robust unsupervised behavioral classification

Unsupervised behavioral classification approaches enable the unbiased quantification of animal behavior by processing data features—image pixel intensities (*Berman et al., 2014*; *Cande et al., 2018*), limb markers (*Todd et al., 2017*), or 2D pose (*Pereira et al., 2019*)—to cluster similar behavioral epochs without user intervention and to automatically distinguish between otherwise similar actions. However, with this sensitivity may come a susceptibility to features unrelated to behavior including changes in image size or perspective resulting from differences in camera angle across experimental systems, variable mounting of tethered animals, and inter-animal morphological variability. In theory, each of these issues can be overcome—providing scale and rotational invariance—by using 3D joint angles rather than 2D pose for unsupervised embedding.

To test this possibility, we performed unsupervised behavioral classification (*Figure 4* and *Figure 5*) on video data taken during optogenetic stimulation experiments that repeatedly and reliably drove certain behaviors. Specifically, we optically activated CsChrimson (*Klapoetke et al., 2014*) to elicit backward walking in MDN>CsChrimson animals (*Figure 5—video 1*) (*Bidaye et al., 2014*), or antennal grooming in aDN>CsChrimson animals (*Figure 5—video 2*) (*Hampel et al., 2015*). We also stimulated control animals lacking the UAS-CsChrimson transgene (*Figure 5—video 3*) (MDN-GAL4/+ and aDN-GAL4/+). First, we performed unsupervised behavioral classification using 2D pose data from three adjacent cameras containing keypoints for three limbs on one side of the body. Using these data, we generated a behavioral map (*Figure 4A*). In this map each individual cluster would ideally represent a single behavior (e.g.

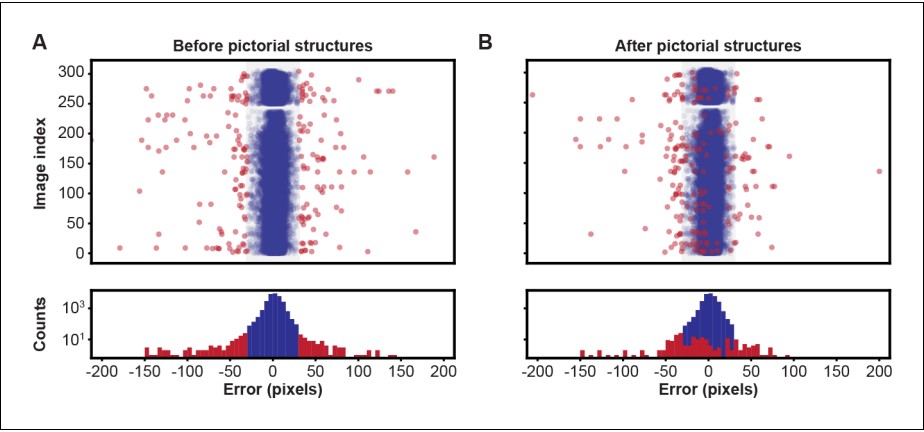

**Figure 3.** Pose estimation accuracy before and after using pictorial structures. Pixel-wise 2D pose errors/residuals (top) and their respective distributions (bottom) (A) before, or (B) after applying pictorial structures. Residuals larger than 35 pixels (red circles) represent incorrect keypoint detections. Those below this threshold (blue circles) represent correct keypoint detections.

DOI: https://doi.org/10.7554/eLife.48571.005

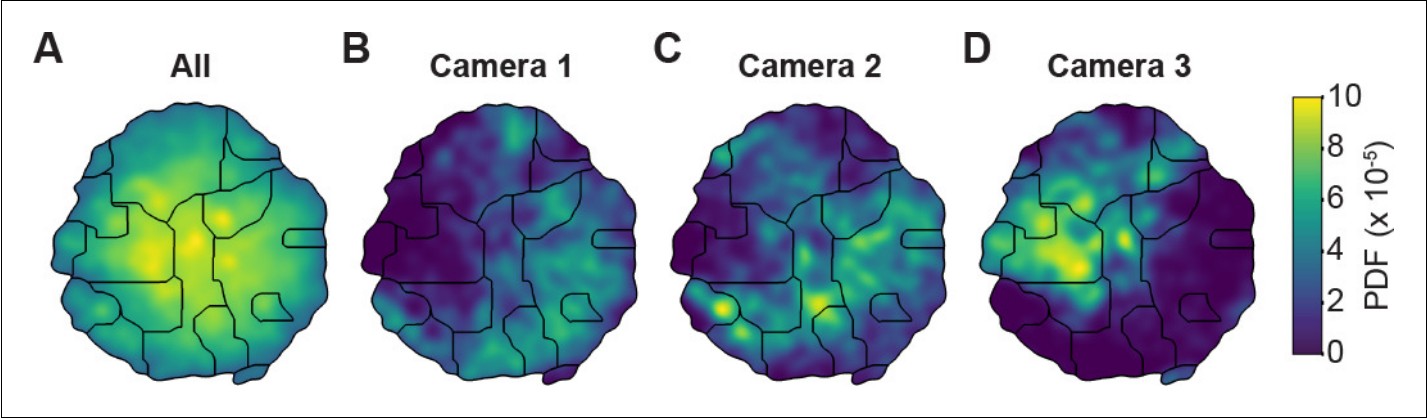

**Figure 4.** Unsupervised behavioral classification of 2D pose data is sensitive to viewing angle. (A) Behavioral map derived using 2D pose data from three adjacent cameras (Cameras 1, 2, and 3) but the same animals and experimental time points. Shown are clusters (black outlines) that are enriched (yellow), or sparsely populated (blue) with data. Different clusters are enriched for data from either (B) camera 1, (C) camera 2, or (D) camera 3. Behavioral embeddings were derived using 1 million frames during 4 s of optogenetic stimulation of MDN>CsChrimson (n = 6 flies, n = 29 trials), aDN>CsChrimson (n = 6 flies, n = 30 trials), and wild-type control animals (MDN-GAL4/+: n = 4 flies, n = 20 trials. aDN-GAL4/+: n = 4 flies, n = 23 trials).

DOI: https://doi.org/10.7554/eLife.48571.006

backward walking, or grooming) and be populated by nearly equal amounts of data from each of the three cameras. This was not the case: data from each camera covered non-overlapping regions and clusters (*Figure 4B–D*). This effect was most pronounced when comparing regions populated by cameras 1 and 2 versus camera 3. Therefore, because the underlying behaviors were otherwise identical (data across cameras were from the same animals and experimental time points), we can conclude that unsupervised behavioral classification of 2D pose data is sensitive to being corrupted by viewing angle differences.

By contrast, performing unsupervised behavioral classification using DeepFly3D-derived 3D joint angles resulted in a map (*Figure 5*) with a clear segregation and enrichment of clusters for different GAL4 driver lines and their associated behaviors, i.e. backward walking (*Figure 5—video 4*), grooming (*Figure 5—video 5*), and forward walking (*Figure 5—video 6*). Thus, 3D pose overcomes serious issues arising from unsupervised embedding of 2D pose data, enabling more reliable and robust behavioral data analysis.

## Discussion

We have developed DeepFly3D, a deep learning-based 3D pose estimation system that is optimized for quantifying limb and appendage movements in tethered, behaving *Drosophila*. By using multiple synchronized cameras and exploiting multiview redundancy, our software delivers robust and accurate pose estimation at the sub-millimeter scale. Ultimately, we may work solely with monocular images by lifting the 2D detections (*Pavlakos et al., 2017b*) to 3D or by directly regressing to 3D (*Tekin et al., 2017*) as has been achieved in human pose estimation studies. Our approach relies on supervised deep learning to train a neural network that detects 2D joint locations in individual camera images. Importantly, our network becomes increasingly competent as it runs: By leveraging the redundancy inherent to a multiple-camera setup, we iteratively reproject 3D pose to automatically detect and correct 2D errors, and then use these corrections to further train the network without user intervention.

None of the techniques we have put together—an approach for multiple-camera calibration that uses the animal itself rather than an external apparatus, an iterative approach to inferring 3D pose using graphical models as well as optimization based on dynamic programming and belief propagation, and a graphical user interface and active learning policy for interacting with, annotating, and correcting 3D pose data—are fly-specific. They could easily be adapted to other limbed animals, from mice to primates and humans. The only thing that would have to change significantly are the

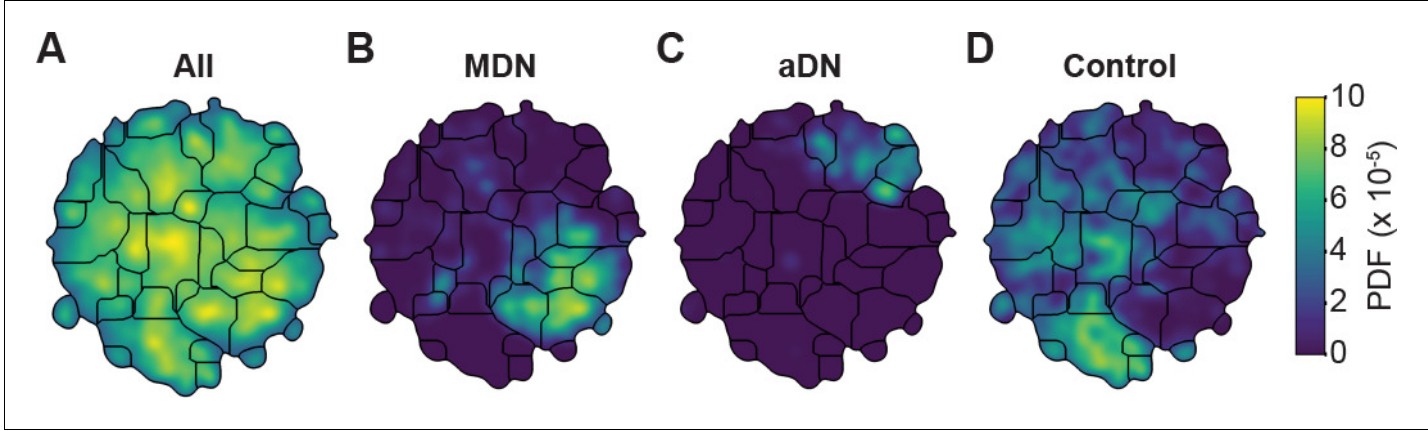

**Figure 5.** Unsupervised behavioral classification of 3D joint angle data. Behavioral embeddings were calculated using 3D joint angles from the same 1 million frames used in *Figure 4A*. (**A**) Behavioral map combining all data during 4 s of optogenetic stimulation of MDN>CsChrimson (n = 6 flies, n = 29 trials), aDN>CsChrimson (n = 6 flies, n = 30 trials), and wild-type control animals (For MDN-Gal4/+, n = 4 flies, n = 20 trials. For aDN-Gal4/+ n = 4 flies, n = 23 trials). The same behavioral map is shown with only the data from (**B**) MDN>CsChrimson stimulation, (**C**) aDN>CsChrimson stimulation, or (**D**) control animal stimulation. Associated videos reveal that these distinct map regions are enriched for backward walking, antennal grooming, and forward walking, respectively.

DOI: https://doi.org/10.7554/eLife.48571.007

The following videos are available for figure 5:

**Figure 5—video 1.** Representative MDN>CsChrimson optogenetically activated backward walking. Orange circle indicates LED illumination and CsChrimson activation.

DOI: https://doi.org/10.7554/eLife.48571.008

**Figure 5—video 2.** Representative aDN>CsChrimson optogenetically activated antennal grooming. Orange circle indicates LED illumination and CsChrimson activation.

DOI: https://doi.org/10.7554/eLife.48571.009

**Figure 5—video 3.** Representative control animal behavior during illumination. Orange circle indicates LED illumination and CsChrimson activation.

DOI: https://doi.org/10.7554/eLife.48571.010

**Figure 5—video 4.** Sample behaviors from 3D pose cluster enriched in backward walking.

DOI: https://doi.org/10.7554/eLife.48571.011

**Figure 5—video 5.** Sample behaviors from 3D pose cluster enriched in antennal grooming.

DOI: https://doi.org/10.7554/eLife.48571.012

**Figure 5—video 6.** Sample behaviors from 3D pose cluster enriched in forward walking.

DOI: https://doi.org/10.7554/eLife.48571.013

dimensions of the experimental setup. This would remove the need to deal with the very small scales *Drosophila* requires and would, in practice, make pose estimation easier. In the Materials and methods section, we explain in detail how organism-specific features of DeepFly3D— bone segment length, number of legs, and camera focal distance—can be modified to study, for example, humans, primates, rodents, or other insects.

As in the past, we anticipate that the development of new technologies for quantifying behavior will open new avenues and enhance existing lines of investigation. For example, deriving 3D pose using DeepFly3D can improve the resolution of studies examining how neuronal stimulation influences animal behavior (*Cande et al., 2018*; *McKellar et al., 2019*), the precision and predictive power of efforts to define natural action sequences (*Seeds et al., 2014*; *McKellar et al., 2019*), the assessment of interventions that target models of human disease (*Feany and Bender, 2000*; *Hewitt and Whitworth, 2017*), and links between neural activity and animal behavior—when coupled with recording technologies like 2-photon microscopy (*Seelig et al., 2010*; *Chen et al., 2018*). Importantly, 3D pose improves the robustness of unsupervised behavioral classification approaches. Therefore, DeepFly3D is a critical step toward the ultimate goal of achieving fully-automated, high-fidelity behavioral data analysis.

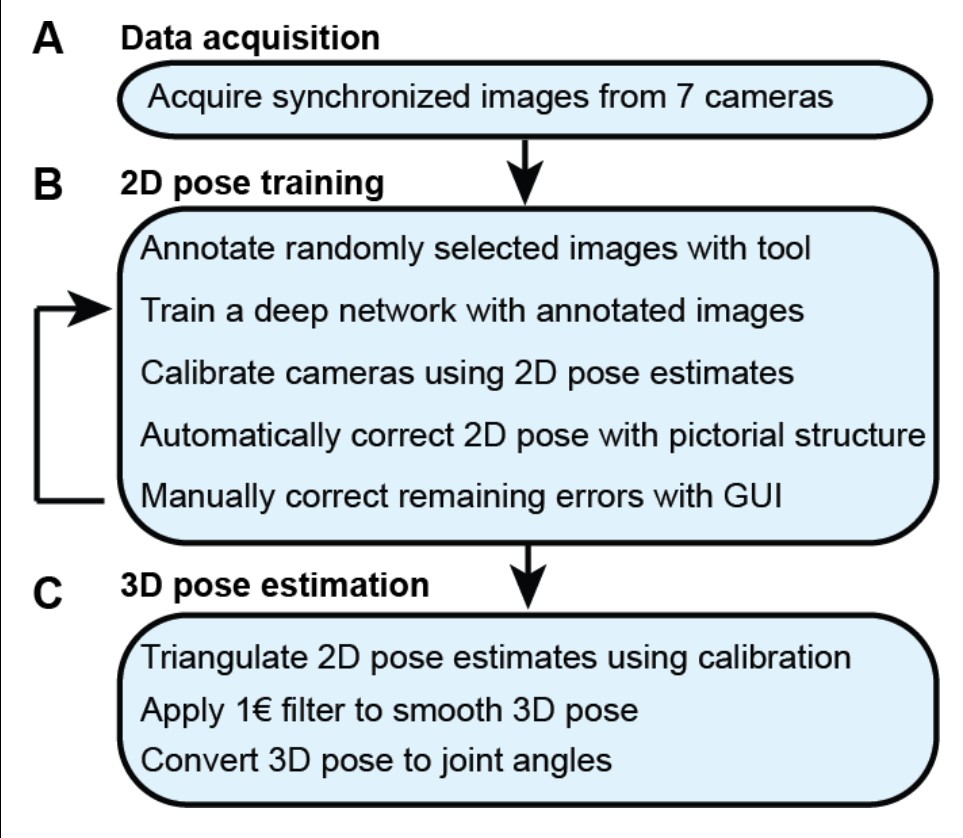

**Figure 6.** The DeepFly3D pose estimation pipeline. (**A**) Data acquisition from the multi-camera system. (**B**) Training and retraining of 2D pose. (**C**) 3D pose estimation.

DOI: https://doi.org/10.7554/eLife.48571.014

## Materials and methods

With synchronized *Drosophila* video sequences from seven cameras in hand, the first task for Deep-Fly3D is to detect the 2D location of 38 landmarks. These 2D locations of the same landmarks seen across multiple views are then triangulated to generate 3D pose estimates. This pipeline is depicted in *Figure 6*. First, we will describe our deep learning-based approach to detect landmarks in images. Then, we will explain the triangulation process that yields full 3D trajectories. Finally, we will describe how we identify and correct erroneous 2D detections automatically.

### 2D pose estimation

#### Deep network architecture

We aim to detect five joints on each limb, six on the abdomen, and one on each antenna, giving a total of 38 keypoints per time instance. To achieve this, we adapted a state-of-the-art Stacked Hourglass human pose estimation network (*Newell et al., 2016*) by changing the input and output layers to accommodate a new input image resolution and a different number of tracked points. A single hourglass stack consists of residual bottleneck modules with max pooling, followed by up-sampling layers and skip connections. The first hourglass network begins with a convolutional layer and a pooling layer to reduce the input image size from $256 \times 512$ to $64 \times 128$ pixels. The remaining hourglass input and output tensors are $64 \times 128$. We used 8 stacks of hourglasses in our final implementation. The output of the network is a stack of probability maps, also known as heatmaps or confidence maps. Each probability map encodes the location of one keypoint, as the belief of the network that a given pixel contains that particular tracked point. However, probability maps do not formally define a probability distribution; their sum over all pixels does not equal 1.

## 2D pose training dataset

We trained our network for 19 keypoints, resulting in the tracking of 38 points when both sides of the fly are taken into account. Determining which images to use for training purposes is critical. The intuitively simple approach—training with randomly selected images—may lead to only marginal improvements in overall network performance. This is because images for which network predictions can already be correctly made give rise to only small gradients during training. On the other hand, manually identifying images that may lead to incorrect network predictions is highly laborious. Therefore, to identify such challenging images, we exploited the redundancy of having multiple camera views (see section *3D pose correction*). Outliers in individual camera images were corrected automatically using images from other cameras, and frames that still exhibited large reprojection errors on multiple camera views were selected for manual annotation and network retraining. This combination of self supervision and active learning permits faster training using a smaller manually annotated dataset (*Simon et al., 2017*). The full annotation and iterative training pipeline is illustrated in *Figure 6*. In total, 40,063 images were annotated: 5,063 were labeled manually in the first iteration, 29,000 by automatic correction, and 6,000 by manually correcting those proposed by the active learning strategy.

## Deep network training procedure

We trained our Stacked Hourglass network to regress from $256 \times 512$ pixel grayscale video images to multiple $64 \times 128$ probability maps. Specifically, during training and testing, networks output a $19 \times 64 \times 128$ tensor; one $64 \times 128$ probability map per tracked point. During training, we created probability maps by embedding a 2D Gaussian with mean at the ground-truth point and 1px symmetrical extent (i.e. $\sigma = 1px$) on the diagonal of the covariance matrix. We calculated the loss as the $L_2$ distance between the ground-truth and predicted probability maps. During testing, the final network prediction for a given point was the probability map pixel with maximum probability. We started with a learning rate of 0.0001 and then multiplied the learning rate by a factor of 0.1 once the loss function plateaued for more than five epochs. We used an RMSPROP optimizer for gradient descent, following the original Stacked Hourglass implementation, with a batch-size of eight images. Using 37,000 training images, the Stacked Hourglass network usually converges to a local minimum after 100 epochs (20 h on a single GPU).

## Network training details

Variations in each fly's position across experiments are handled by the translational invariance of the convolution operation. In addition, we artificially augment training images to improve network generalization for further image variables. These variables include (i) illumination conditions – we randomly changed the brightness of images using a gamma transformation, (ii) scale – we randomly rescaled images between 0.80x - 1.20x, and (iii) rotation – we randomly rotated images and corresponding probability maps ±15°. This augmentation was enough to compensate for real differences in the size and orientation of tethered flies across experiments. Furthermore, as per general practice, the mean channel intensity was subtracted from each input image to distribute annotations symmetrically around zero. We began network training using pretrained weights from the MPII human pose dataset (*Andriluka et al., 2014*). This dataset consists of more than 25,000 images with 40,000 annotations, possibly with multiple ground-truth human pose labels per image. Starting with a pretrained network results in faster convergence. However, in our experience, this does not affect final network accuracy in cases with a large amount of training data. We split the dataset into 37,000 training images, 2,063 testing images, and 1,000 validation images. None of these subsets shared common images or common animals, to ensure that the network could generalize across animals, and experimental setups. 5,063 of our training images were manually annotated, and the remaining data were automatically collected using belief propagation, graphical models, and active learning, (see section *3D pose correction*). Deep neural network parameters need to be trained on a dataset with manually annotated ground-truth key point positions. To initialize the network, we collected annotations using a custom multicamera annotation tool that we implemented in JavaScript using Google Firebase (*Figure 7*). The DeepFly3D annotation tool operates on a simple web-server, easing the distribution of annotations across users and making these annotations much easier to inspect and control.

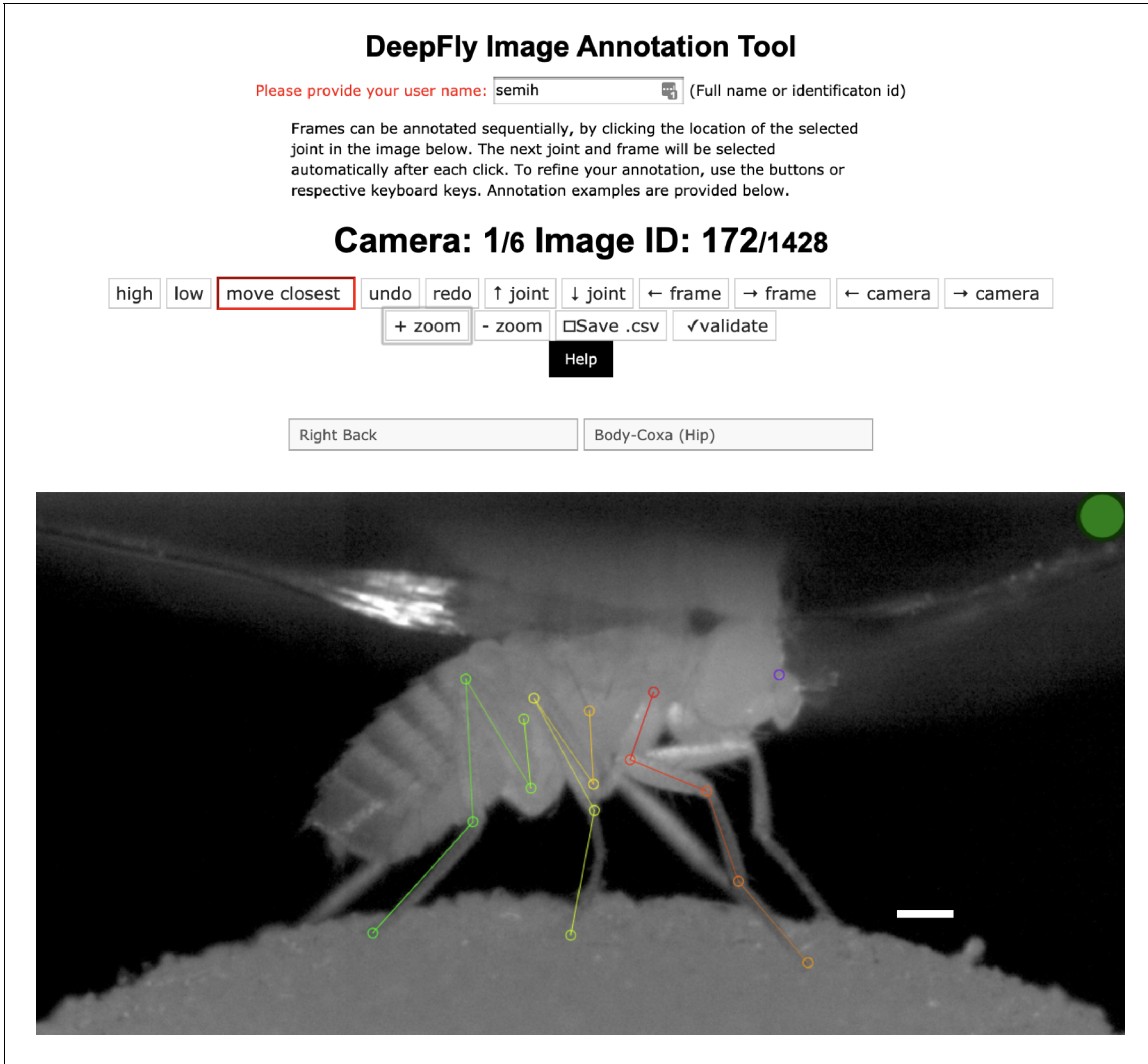

**Figure 7.** The DeepFly3D annotation tool. This GUI allows the user to manually annotate joint positions on images from each of seven cameras. Because this tool can be accessed from a web browser, annotations can be performed in a distributed manner across multiple users more easily. A full description of the annotation tool can be found in the online documentation: https://github.com/NeLy-EPFL/DeepFly3D. Scale bar is 50 pixels.
DOI: https://doi.org/10.7554/eLife.48571.015

## Computing hardware and software

We trained our model on a desktop computing workstation running on an Intel Core i9-7900X CPU, 32 GB of DDR4 RAM, and a GeForce GTX 1080. With 37,000 manually and automatically labeled images, training takes nearly 20 h on a single GeForce GTX 1080 GPU. Our code is implemented with Python 3.6, Pytorch 0.4 and CUDA 9.2. Using this desktop configuration, our network can run at 100 Frames-Per-Second (FPS) using the 8-stack variant of the Hourglass network, and can run at 420 FPS using the smaller 2-stack version. Thanks to an effective initialization step, calibration takes 3–4 s. Error checking and error correction can be performed at 100 FPS and 10 FPS, respectively. Error correction is only performed in response to large reprojection errors and does not create a bottleneck in the overall speed of the pipeline.

## Accuracy analysis

Consistent with the human pose estimation literature, we report accuracy as Percentage of Correct Keypoints (PCK) and Root Mean Squared Error (RMSE). PCK refers to the percentage of detected points lying within a specific radius from the ground-truth label. We set this threshold as 50 pixels,

which is roughly one third of the 3D length of the femur. The final estimated position of each key-point was obtained by selecting the pixel with the largest probability value on the relevant probability map. We compared DeepFly3D's annotations with manually annotated ground-truth labels to test our model's accuracy. For RMSE, we report the square root of average pixel distance between the prediction and the ground-truth location of the tracked point. We remove trivial points such as the body-coxa and coxa-femur—which remain relatively stationary—to fairly evaluate our algorithms and to prevent these points from dominating our accuracy measurements.

## From 2D landmarks to 3D trajectories

In the previous section, we described our approach to detect 38 2D landmarks. Let $\mathbf{x}_{c,j} \in \mathbb{R}^2$ denote the 2D position of landmark $j$ in the image acquired by camera $c$. For each landmark, our task is now to estimate the corresponding 3D position, $\mathbf{X}_j \in \mathbb{R}^3$. To accomplish this, we used triangulation and bundle-adjustment (*Hartley and Zisserman, 2000*) to compute 3D locations, and we used pictorial structures (*Felzenszwalb and Huttenlocher, 2005*) to enforce geometric consistency and to eliminate potential errors caused by misdetections. We present these steps below.

### Pinhole camera model

The first step is to model the projection operation that relates a specific $\mathbf{X}_j$ to its seven projections in each camera view $\mathbf{x}_{c,j}$. To make this easier, we follow standard practice and convert all Cartesian coordinates $[x_c, y_c, z_c]$ to homogeneous ones $[x_h, y_h, z_h, s]$ such that $x_c = x_h/s$, $y_c = y_h/s$, $z_c = z_h/s$. From now on, we will assume that all points are expressed in homogeneous coordinates and omit the $h$ subscript. Assuming that these coordinates are expressed in a coordinate system whose origin is in the optical center of the camera and whose z-axis is its optical axis, the 2D image projection $[u, v]$ of a 3D homogeneous point $[x, y, z, 1]$ can be written as

$$u = U/W\,,$$
$$v = V/W\,,$$
$$\begin{bmatrix} U \\ V \\ W \end{bmatrix} = \mathbf{K} \begin{bmatrix} x \\ y \\ z \\ 1 \end{bmatrix}\,, \text{ with } \mathbf{K} = \begin{bmatrix} f_x & 0 & c_x & 0 \\ 0 & f_y & c_y & 0 \\ 0 & 0 & 1 & 0 \end{bmatrix}\,, \tag{1}$$

where the $3 \times 4$ matrix $\mathbf{K}$ is known as the *intrinsic parameters matrix*—scaling in the $x$ and $y$ direction and image coordinates of the principal point $c_x$ and $c_y$—that characterizes the camera settings.

In practice, the 3D points are not expressed in a camera fixed coordinate system, especially in our application where we use seven different cameras. Therefore, we use a world coordinate system that is common to all cameras. For each camera, we must therefore convert 3D coordinates expressed in this world coordinate system to camera coordinates. This requires rotating and translating the coordinates to account for the position of the camera's optical center and its orientation. When using homogeneous coordinates, this is accomplished by multiplying the coordinate vector by a $4 \times 4$ *extrinsic parameters matrix*

$$\mathbf{M} = \begin{bmatrix} \mathbf{R} & \mathbf{T} \\ 0 & 1 \end{bmatrix}\,, \tag{2}$$

where $\mathbf{R}$ is a $3 \times 3$ rotation matrix and $\mathbf{T}$ a $3 \times 1$ translation vector. Combining *Equation 1* and *Equation 2* yields

$$u = U/W\,,$$
$$v = V/W\,,$$
$$\begin{bmatrix} U \\ V \\ W \end{bmatrix} = \mathbf{P} \begin{bmatrix} x \\ y \\ z \\ 1 \end{bmatrix}\,, \tag{3}$$

where $\mathbf{P} = \mathbf{MK}$ is a $3 \times 4$ matrix.

## Camera distortion

The pinhole camera model described above is an idealized one. The projections of real cameras deviate from it. These deviations are referred to as distortions and must be accounted for. The most

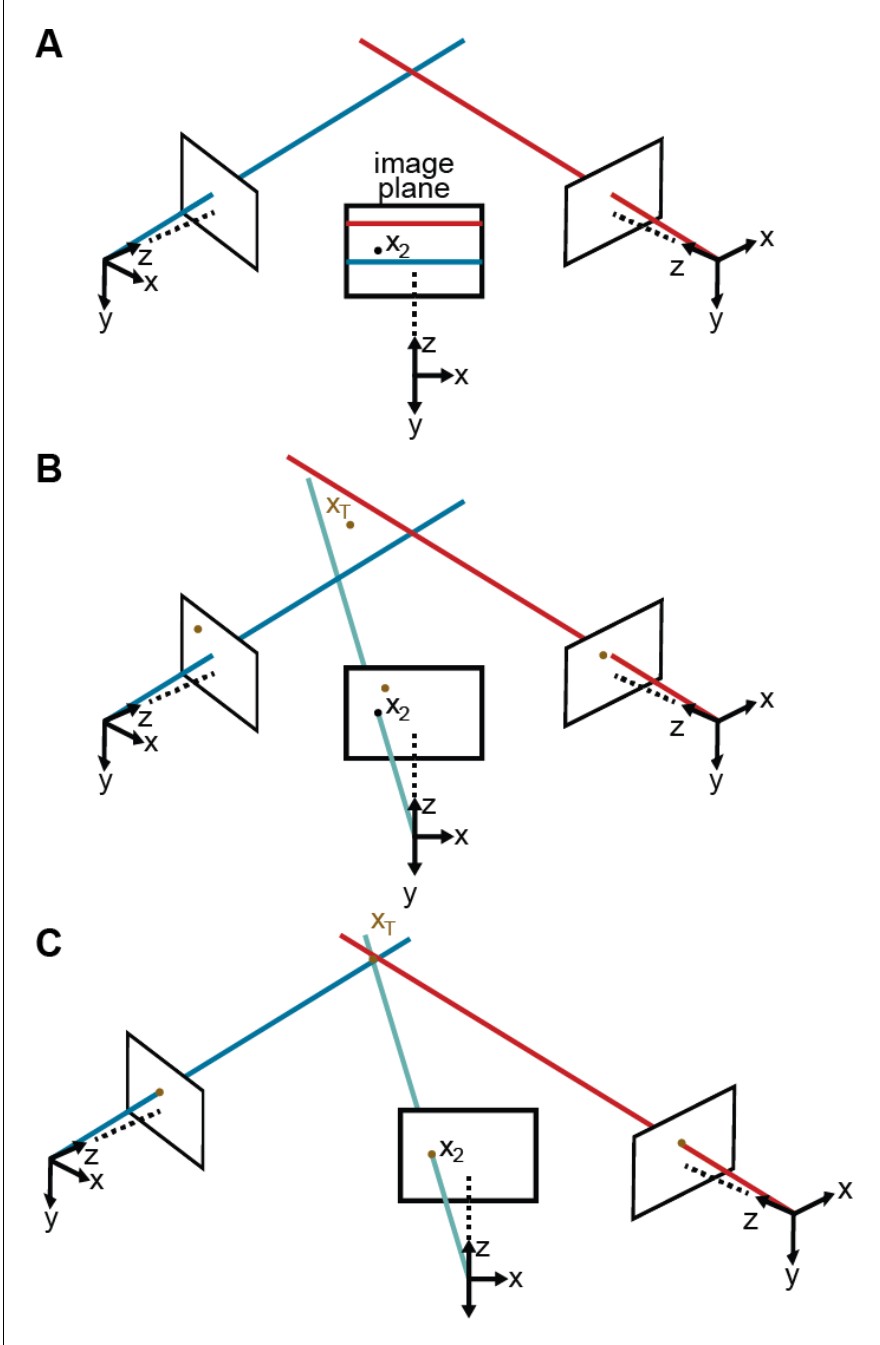

**Figure 8.** Camera calibration. (A) Correcting erroneous 2D pose estimations by using epipolar relationships. Only 2D pose estimates without large epipolar errors are used for calibration. $x_2$ represents a 2D pose estimate from the middle camera. Epipolar lines are indicated as blue and red lines on the image plane. (B) The triangulated point, $X_T$, uses the initial camera parameters. However, due to the coarse initialization of each camera's extrinsic properties, observations from each camera do not agree with one another and do not yield a reasonable 3D position estimate. (C) The camera locations are corrected, generating an accurate 3D position estimate by optimizing *Equation 7* using only the pruned 2D points.
DOI: https://doi.org/10.7554/eLife.48571.016

significant distortion is known as radial distortion because the error grows with the distance from the image center. For the cameras we use, radial distortion can be expressed as

$$u_{\text{pinhole}} = u\left(1 + k_1^x r^2 + k_2^x r^4\right),$$
$$v_{\text{pinhole}} = v\left(1 + k_1^y r^2 + k_2^y r^4\right),$$

(4)

where $[u,v]$ is the actual projection of a 3D point and $[u_{\text{pinhole}}, v_{\text{pinhole}}]$ is the one the pinhole model predicts. In other words, the four parameters $\{k_1^x, k_2^x, k_1^y, k_2^y\}$ characterize the distortion. From now on, we will therefore write the full projection as

$$\mathbf{X} = \pi(\mathbf{x}) = f_d(f_p(\mathbf{x})),$$
$$\mathbf{X} = [x,y,z],$$
$$\mathbf{X} = [u,v],$$

(5)

where $f_p$ denotes the ideal pinhole projection of *Equation 3* and $f_d$ the correction of *Equation 4*.

## Triangulation

We can associate to each of the seven cameras a projection function $\pi_c$ like the one in *Equation 5*, where $c$ is the camera number. Given a 3D point and its projections $\mathbf{x}_c$ in the images, its 3D coordinates can be estimated by minimizing the *reprojection error*

$$\underset{\mathbf{X} \in \mathbb{R}^4}{\operatorname{argmin}} \sum_{c=1}^{7} e_c \|\pi_c(\mathbf{X}) - \mathbf{x}_c\|^2,$$

(6)

where $e_c$ is one if the point was visible in image $c$ and zero otherwise. In the absence of camera distortion, that is, when the projection $\pi$ is a purely linear operation in homogeneous coordinates, this can be done for any number of cameras by solving a Singular Value Decomposition (SVD) problem (*Hartley and Zisserman, 2000*). In the presence of distortions, we replace the observed $u$ and $v$ coordinates of the projections by the corresponding $u_{\text{pinhole}}$ and $u_{\text{pinhole}}$ values of *Equation 5* before performing the SVD.

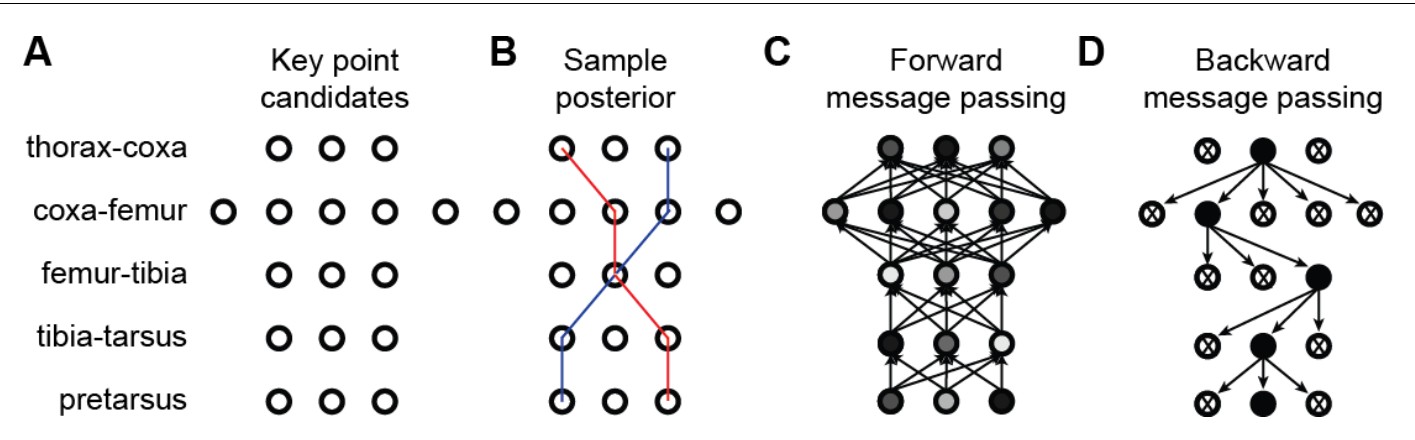

**Figure 9.** 3D pose correction for one leg using the MAP solution and pictorial structures. (A) Candidate 3D pose estimates for each keypoint are created by triangulating local maxima from probability maps generated by the Stacked Hourglass deep network. (B) For a selection of these candidate estimates, we can assign a probability using *Equation 8*. However, calculating this probability for each pair of points is computationally intractable. (C) By exploiting the chain structure of *Equation 8*, we can instead pass a probability distribution across layers using a belief propagation algorithm. Messages are passed between layers as a function of parent nodes, describing the belief of the child nodes on each parent node. Grayscale colors represent the calculated belief of each node where darker colors indicate higher belief. (D) Corrected pose estimates are obtained during the second backward iteration, by selecting the nodes with largest belief. We discard nodes (x's) that have non-maximal belief during backwards message passing. Note that beliefs have been adjusted after forward message passing.
DOI: https://doi.org/10.7554/eLife.48571.017

## Camera calibration

Triangulating as described above requires knowing the projection matrices $\mathbf{P}_c$ of *Equation 3* for each camera $c$, corresponding distortion parameters $\{k_1^x, k_2^x, k_1^y, k_2^y\}$ of *Equation 4*, together with the intrinsic parameters of focal length and principal point offset. In practice, we use the focal length and principal point offset provided by the manufacturer and estimate the remaining parameters automatically: the three translations and three rotations for each camera that define the corresponding matrix $\mathbf{M}$ of extrinsic parameters along with the distortion parameters.

To avoid having to design the exceedingly small calibration pattern that more traditional methods use to estimate these parameters, we use the fly itself as calibration pattern and minimize the reprojection error of *Equation 6* for all joints simultaneously while allowing the camera parameters to also change. In other words we look for

$$
\underset{\substack{\boldsymbol{\pi}_{c_{1 \leq c \leq 7}} \\ \mathbf{X}_{j_{1 \leq j \leq m}}}}{\operatorname{argmin}} \sum_{c=1}^{7} \sum_{j=1}^{m} e_{c,j} \rho(\boldsymbol{\pi}_c(\mathbf{X}_j) - \mathbf{x}_{c,j}), \tag{7}
$$

where $\mathbf{X}_j$ and $\mathbf{x}_{c,j}$ are the 3D locations and 2D projections of the landmarks introduced above and $\rho$ denotes the Huber loss. *Equation 7* is known as bundle-adjustment (*Hartley and Zisserman, 2000*). Huber loss is defined as

$$
\rho_\delta(a) = \begin{cases} \frac{1}{2}a^2 & \text{for} |a| \leq \delta \\ \delta(|a| - \frac{1}{2}\delta) & \text{otherwise} \end{cases}.
$$

Replacing the squared loss by the Huber loss makes our approach more robust to erroneous detections $\mathbf{x}_{c,j}$. We empirically set $\delta$ to 20 pixels. Note that we perform this minimization with respect to ten degrees-of-freedom per camera: three translations, three rotations, and four distortions.

For this optimization to work properly, we need to initialize these 10 parameters and we need to reduce the number of outliers. To achieve this, the initial distortion parameters are set to zero. We also produce initial estimates for the three rotation and three translation parameters by measuring the distances between adjacent cameras and their relative orientations. To initialize the rotation and translation vectors, we measure the distance and the angle between adjacent cameras, from which we infer rough initial estimates. Finally, we rely on epipolar geometry (*Hartley and Zisserman, 2000*) to automate outlier rejection. Because the cameras form a rough circle and look inward, the epipolar lines are close to being horizontal *Figure 8A*. Thus, corresponding 2D projections must belong to the same image rows, or at most a few pixels higher or lower. In practice, this means checking if all 2D predictions lie in nearly the same rows and discarding a priori those that do not.

## 3D pose correction

The triangulation procedure described above can produce erroneous results when the 2D estimates of landmarks are wrong. Additionally, it may result in implausible 3D poses for the entire animal because it treats each joint independently. To enforce more global geometric constraints, we rely on pictorial structures (*Felzenszwalb and Huttenlocher, 2005*) as described in *Figure 9*. Pictorial structures encode the relationship between a set of variables (in this case the 3D location of separate tracked points) in a probabilistic setting using a graphical model. This makes it possible to consider multiple 2D locations $\mathbf{x}_{c,j}$ for each landmark $\mathbf{X}_c$ instead of only one. This increases the likelihood of finding the true 3D pose.

### Generating multiple candidates

Instead of selecting landmarks as the locations with the maximum probability in maps output by our Stacked Hourglass network, we generate multiple candidate 2D landmark locations $x_{c,j}$. From each probability map, we select 10 local probability maxima that are at least one pixel apart from one another. Then, we generate 3D candidates by triangulating 2D candidates in every tuple of cameras. Because a single point is visible from at most four cameras, this results in at most $\binom{4}{2} \times 10^2$ candidates for each tracked point.

## Choosing the best candidates

To identify the best subset of resulting 3D locations, we introduce the probability distribution $P(L|I, \theta)$ that assigns a probability to each solution $L$, consisting of 38 sets of 2D points observed from each camera. Our goal is then to find the most likely one. More formally, $P$ represents the likelihood of a set of tracked points $L$, given the images, model parameters, camera calibration, and geometric constraints. In our formulation, $I$ denotes the seven camera images $I = \{I_c\}_{1 \leq c \leq 7}$ and $\theta$ represents the set of projection functions $\pi_c$ for camera $c$ along with a set of length distributions $S_{i,j}$ between each pair of points $i$ and $j$ that are connected by a limb. $L$ consists of a set of tracked points $\{L_i\}_{1 \leq i \leq n}$, where each $L_i$ describes a set of 2D observations $l_{i,c}$ from multiple camera views. These are used to triangulate the corresponding 3D point locations $\bar{l}_i$. If the set of 2D observations is incomplete, as some points are totally occluded in some camera views, we triangulate the 3D point $\bar{l}_i$ using the available ones and replace the missing observations by projecting the recovered 3D positions into the images, $\pi_c(\bar{l}_i)$ in *Equation 3*. In the end, we aim to find the solution $\hat{L} = argmax_L P(L|I, \theta)$. This is known as Maximum a Posteriori (MAP) estimation. Using Bayes rule, we write

$$P(L|I, \theta) \propto P(I|L, \theta)P(L|\theta) \,, \tag{8}$$

where the two terms can be computed separately. We compute $P(I|J, \theta)$ using the probability maps $H_{j,c}$ generated by the Stacked Hourglass network for the tracked point $j$ for camera $c$. For a single joint $j$ seen by camera $c$, we model the likelihood of observing that particular point using $P(H_{j,c}|l_{j,c})$, which can be directly read from the probability maps as the pixel intensity. Ignoring the dependency between the cameras, we write the overall likelihood as the product of the individual likelihood terms

$$P(I|L, \theta) = P(H|L) \propto \prod_{i=1}^{n} \prod_{c=1}^{7} P(H_{j,c}|l_{i,c}) \,, \tag{9}$$

which can be read directly from the probability maps as pixel intensities and represent the network's confidence that a particular keypoint is located at a particular pixel. When a point is not visible from a particular camera, we assume the probability map only contains a constant non-zero probability, which does not affect the final solution. We express $P(L|\theta)$ as

$$P(L|\theta) = P(L|\pi, S) = \prod_{(i,j) \in E} P(\bar{l}_i, \bar{l}_j | S_{i,j}) \prod_{j=1}^{n} \prod_{c=1}^{7} e_{c,j} \|\pi_c(\bar{l}_j) - l_{c,j}\|_2^{-1} \,, \tag{10}$$

where pairwise dependencies $P(\bar{l}_i, \bar{l}_j | S_{i,j})$ between two variables respect the segment length constraint when the variables are connected by a limb. The length of segments defined by pairs of connected 3D points follows a normal distribution. Specifically, we model $P(\bar{l}_i, \bar{l}_j | S_{i,j})$ as $S_{i,j}(\bar{l}_i, \bar{l}_j) = \mathcal{N}(\|\bar{l}_i - \bar{l}_j\| - \mu_{i,j}, \sigma_{i,j})$. We model the reprojection error for a particular point $j$ as $\prod_{c=1}^{7} e_{c,j} \|\pi_c(\bar{l}_j) - l_{c,j}\|_2^{-1}$ which is set to zero using the variable $e_{c,j}$ denoting the visibility of the point $j$ from camera $c$. If a 2D observation for a particular camera is manually set by a user with the DeepFly3D GUI, we take it to be the only possible candidate for that particular image and we set $P(L_j|H)$ to 1, where $j$ denotes the manually assigned pixel location.

## Solving the MAP problem using the Max-Sum algorithm

For general graphs, MAP estimation with pairwise dependencies is NP-hard and therefore intractable. However, in the specific case of non-cyclical graphs, it is possible to solve the inference problem using belief propagation (*Bishop, 2006*). Since the fly's skeleton has a root and contains no loops, we can use a message passing approach (*Felzenszwalb and Huttenlocher, 2005*). It is closely related to Viterbi recurrence and propagates the unary probabilities $P(L_j|L_i)$ between the edges of the graph starting from the root and ending at the leaf nodes. This first propagation ends with the computation of the marginal distribution for the leaf node variables. During the subsequent backward iteration, as $P(L_j)$ for leaf node is computed, the point $L_j$ with maximum posterior probability is selected in $O(k)$ time, where $k$ is the upper bound on the number of proposals for a single tracked point. Next, the distribution $P(L_i|L_j)$ is calculated, adjacent nodes for the leaf node. Continuing this

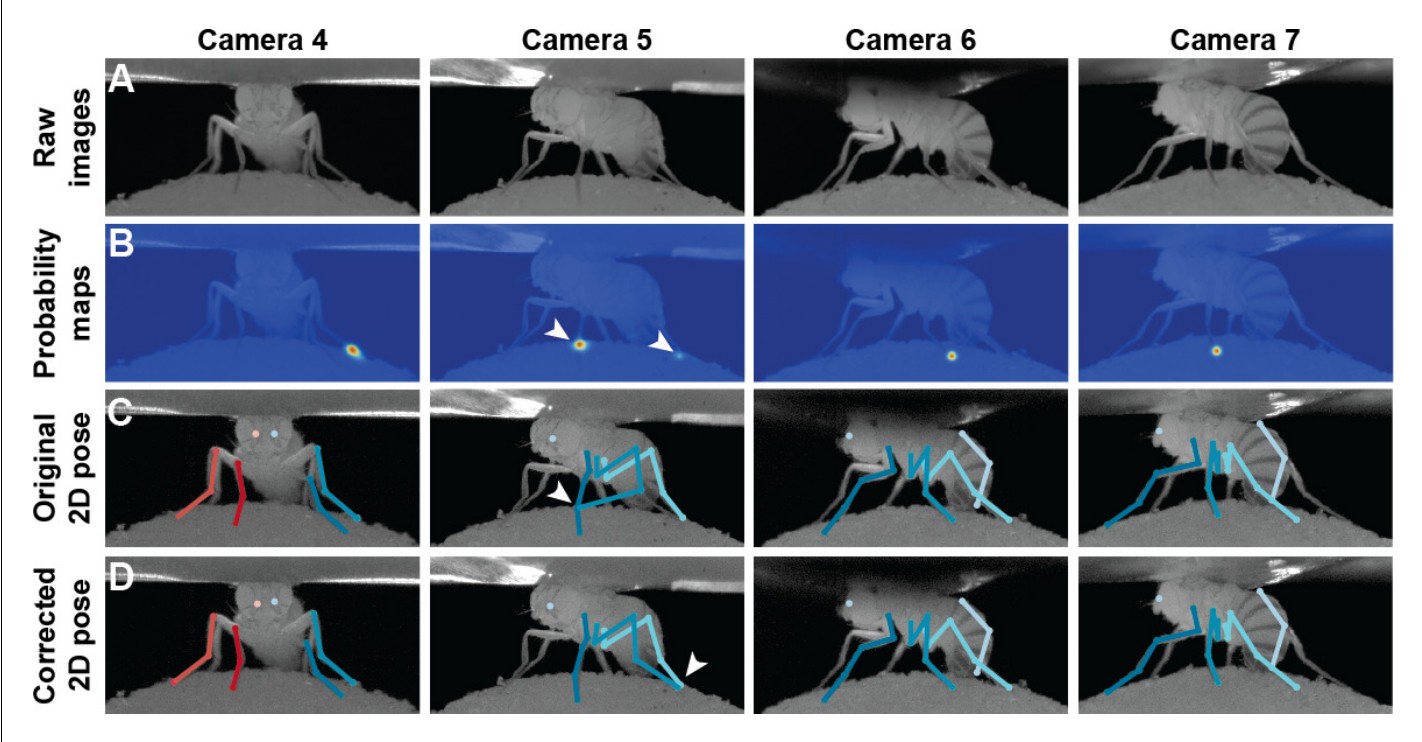

**Figure 10.** Pose correction using pictorial structures. (**A**) Raw input data from four cameras, focusing on the pretarsus of the middle left leg. (**B**) Probability maps for the pretarsus output from the Stacked Hourglass deep network. Two maxima (white arrowheads) are present on the probability maps for camera 5. The false-positive has a larger unary probability. (**C**) Raw predictions of 2D pose estimation without using pictorial structures. The pretarsus label is incorrectly applied (white arrowhead) in camera 5. By contrast, cameras 4, 6, and 7 are correctly labeled. (**D**) Corrected pose estimation using pictorial structures. The false-positive is removed due to the high error measured in *Equation 8*. The newly corrected pretarsus label for camera five is shown (white arrowhead).

DOI: https://doi.org/10.7554/eLife.48571.019

process on all the remaining points results in a MAP solution for the overall distribution $P(L)$, as shown in *Figure 9*, with overall $O(k^2)$ computational complexity.

## Learning the parameters

We learn the parameters for the set of pairwise distributions $S_{i,j}$ using a maximum likelihood process and assuming the distributions to be Gaussian. We model the segment length $S_{i,j}$ as the euclidean distance between the points $\bar{l}_j$ and $\bar{l}_j$. We then solve for $argmax_S P(S|L, \theta)$, assuming segments have a Gaussian distribution resulting from the Gaussian noise in point observations $L$. This gives us the mean and variance, defining each distribution $S_{i,j}$. We exclude the same points that we removed from the calibration procedure, that exhibit high reprojection error.

In practice, we observe a large variance for pretarsus values (*Figure 10*). This is because occlusions occasionally shorten visible tarsal segments. To eliminate the resulting bias, we treat these limbs differently from the others and model the distribution of tibia-tarsus and tarsus-tip points as a Beta distribution, with parameters found using a similar Maximum Likelihood Estimator (MLE) formulation. Assuming the observation errors to be Gaussian and zero-centered, the bundle adjustment procedure can also be understood as an MLE of the calibration parameters (*Triggs et al., 2000*). Therefore, the entire set of parameters for the formulation can be learned using MLE. Thus, prior information about potentially occluded targets can be used to guide inference. For example, in a head-fixed rodent, the left eye may not always be visible from the right-side of the animal. This information can be incorporated into DeepFly3D's inference system in the file, *skeleton.py*, by editing the function *camera_see_joint*. Afterwards, predictions from occluded cameras will not be used to

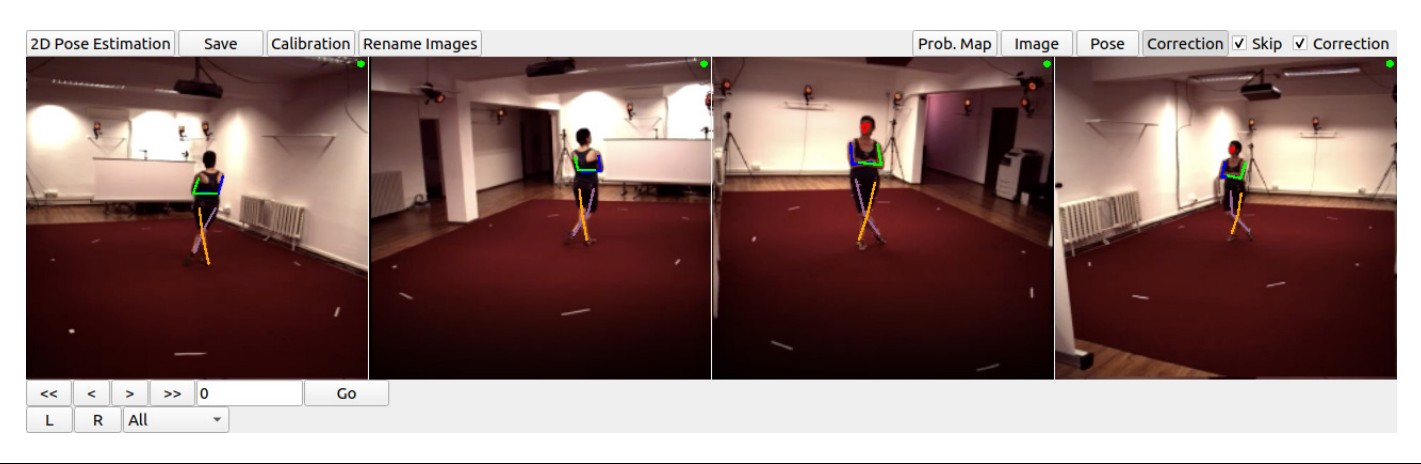

**Figure 11.** DeepFly3D graphical user interface (GUI) applied to with the Human3.6M dataset (*Ionescu et al., 2014*). To use the DeepFly3D GUI on any new dataset (*Drosophila* or otherwise), users can provide an initial small set of manual annotations. Using these annotations, the software calculates the epipolar geometry, performs camera calibration, and trains the 2D pose estimation deep network. A description of how to adopt DeepFly3D for new datasets can be found in the Materials and methods section and, in greater detail, online: https://github.com/NeLy-EPFL/DeepFly3D. This figure is licensed for academic use only and thus is not available under CC-BY and is exempt from the CC-BY 4.0 license.
DOI: https://doi.org/10.7554/eLife.48571.020

triangulate a given 3D point. If no such information is provided, every prediction will be used to triangulate a given 3D point.

The pictorial structure formulation can be further expanded using temporal information, penalizing large movements of a single tracked point between two consecutive frames. However, we abstained from using temporal information more extensively for several reasons. First, temporal dependencies would introduce loops in our pictorial structures, thus making exact inference NP-hard as discussed above. This can be handled using loopy belief propagation algorithms (*Murphy et al., 1999*) but requires multiple message passing rounds, which prevents real-time inference without any theoretical guarantee of optimal inference. Second, the rapidity of *Drosophila* limb movements makes it hard to assign temporal constraints, even with fast video recording. Finally, we empirically observed that the current formulation, enforcing structured poses in a single temporal frame, already eliminates an overwhelming majority of false-positives inferred during the pose estimation stage of the algorithm.

## Modifying DeepFly3D to study other animals

DeepFly3D does not assume a circular camera arrangement or that there is one degree of freedom in the camera network. Therefore, it could easily be adapted for 3D pose estimation in other animals, ranging from rodents to primates and humans. We illustrate this flexibility by using DeepFly3D to capture human 3D pose in the Human 3.6M Dataset (http://vision.imar.ro/human3.6m/description.php) very popular, publicly available computer vision benchmarking dataset generated using four synchronized cameras (*Ionescu et al., 2014*; *Ionescu et al., 2011*) (*Figure 11*).

Generally, for any new dataset, the user first needs to provide an initial set of manual annotations. The user would describe the number of tracked points and their relationships to one another in a python setup file. Then, in a configuration file, the user specifies the number of cameras along with the resolutions of input images and output probability maps. DeepFly3D will then use these initial manual annotations to (i) train the 2D Stacked Hourglass network, (ii) perform camera calibration without an external calibration pattern, (iii) learn the epipolar geometry to perform outlier detection, and (iv) learn the segment length distributions $S_{i,j}$. After this initial bootstrapping, DeepFly3D can be then used with pictorial structures and active learning to iteratively improve pose estimation accuracy.

The initial manual annotations can be performed using the DeepFly3D Annotation GUI. Afterwards, these annotations can be downloaded from the Annotation GUI as a CSV file using the *Save*

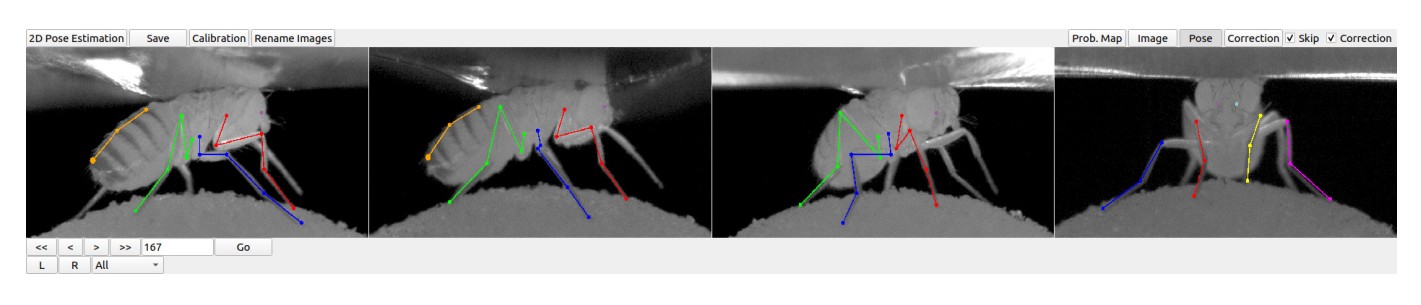

**Figure 12.** DeepFly3D graphical user interface (GUI). The top-left buttons enable operations like 2D pose estimation, camera calibration, and saving the final results. The top-right buttons can be used to visualize the data in different ways: as raw images, probability maps, 2D pose, or the corrected pose following pictorial structures. The bottom-left buttons permit frame-by-frame navigation. A full description of the GUI can be found in the online documentation: https://github.com/NeLy-EPFL/DeepFly3D.
DOI: https://doi.org/10.7554/eLife.48571.018

button (*Figure 7*). Once the CSV file is placed in the images folder, DeepFly3D will automatically read and display the annotations. To train the Stacked Hourglass network, use the *csv-path* flag while running *pose2d.py* (found in *deepfly/pose2d/*). DeepFly3D will then train the Stacked Hourglass network by performing transfer learning using the large MPII dataset and the smaller set of user manual annotations.

To perform camera calibration, the user should select the *Calibration* button on the GUI **Figure 12**. DeepFly3D will then perform bundle adjustment (*Equation 7*) and save the camera parameters in *calibration.pickle* (found in the images folder). The path of this file should then be added to *Config.py* to initialize calibration. These initial calibration parameters will then be used in further experiments for fast and accurate convergence. If the number of annotations is insufficient for accurate calibration, or if bundle adjustment is converging too slowly, an initial rough estimate of the camera locations can be set in *Config.py*. As long as a calibration is set in *Config.py*, DeepFly3D will use it as a projection matrix to calculate the epipolar geometry between cameras. This step is necessary to perform outlier detection on further calibration operations.

DeepFly3D will also learn the distribution $S_{i,j}$, whose non-zero entries are found in *skeleton.py*. One can easily calculate these segment length distribution parameters using the functions provided

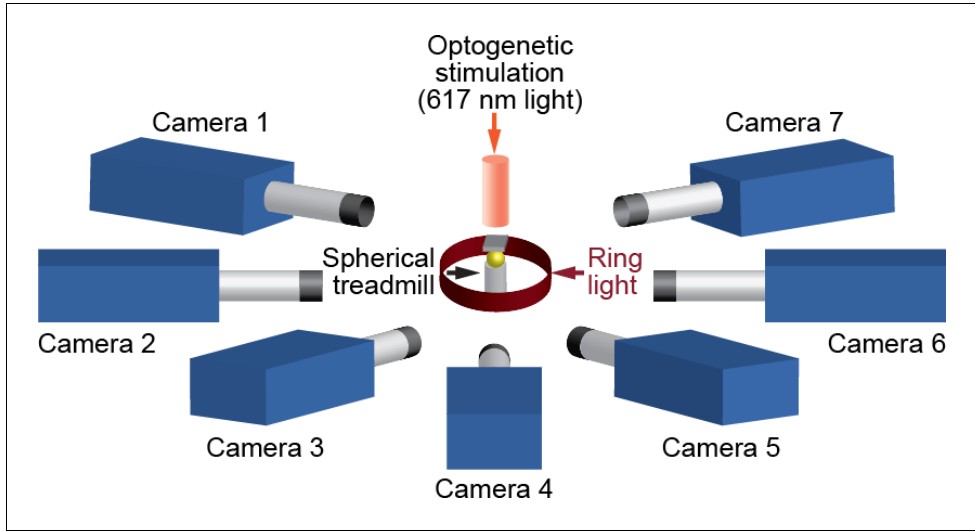

**Figure 13.** A schematic of the seven camera spherical treadmill and optogenetic stimulation system that was used in this study.
DOI: https://doi.org/10.7554/eLife.48571.021

with DeepFly3D. *CameraNetwork* class (found under *deepfly/GUI/*), will then automatically load the points and calibration parameters from the images folder. The function *CameraNetwork.triangulate* will convert 2D annotation points into 3D points using the calibration parameters. The $S_{i,j}$ parameters can then be saved using the *pickle* library (the save path can be set in *Config.py*). The *calcBoneParams* method will then output the segment lengths' mean and variance. These values will then be used with pictorial structures (*Equation 8*).

We provide further technical details for how to adapt DeepFly3D to other multi-view datasets online (https://github.com/NeLy-EPFL/DeepFly3D [*Günel et al., 2019*] copy archived at https://github.com/elifesciences-publications/DeepFly3D).

## Experimental setup

We positioned seven Basler acA1920-155um cameras (FUJIFILM AG, Niederhaslistrasse, Switzerland) 94 mm away from the tethered fly, resulting in a circular camera network with the animal in the center (*Figure 13*). We acquired 960 × 480 pixel video data at 100 FPS under 850 nm infrared ring light illumination (Stemmer Imaging, Pfäffikon Switzerland). Cameras were mounted with 94 mm W.D./ 1.00 x InfiniStix lenses (Infinity Photo-Optical GmbH, Göttingen). Optogenetic stimulation LED light was filtered out using 700 nm longpass optical filters (Edmund Optics, York UK). Each camera's depth of field was increased using 5.8 mm aperture retainers (Infinity Photo-Optical GmbH). To automate the timing of optogenetic LED stimulation and camera acquisition triggering, we use an Arduino (Arduino, Sommerville, MA) and custom software written using the Basler camera API.

We assessed the optimal number of cameras for DeepFly3D and concluded that increasing the number of cameras increases accuracy by stabilizing triangulation. Specifically, we observed the following. (i) Calibration is not a significant source of error: calibrating with fewer than seven cameras does not dramatically increase estimation error. (ii) Having more cameras improves triangulation. Reducing the number of cameras down to four, even having calibrated with seven cameras, results in an increase of 0.05 mm triangulation error. This may be because the camera views are sufficiently different, having largely non-overlapping 2D-detection failure cases. Thus, the redundancy provided by having more cameras mitigates detection errors by finding a 3D pose that is consistent across at least two camera views.

### *Drosophila* transgenic lines

*UAS-CsChrimson* (*Klapoetke et al., 2014*) animals were obtained from the Bloomington Stock Center (Stock #55135). *MDN-1-Gal4* (*Bidaye et al., 2014*) (*VT44845-DBD; VT50660-AD*) was provided by B. Dickson (Janelia Research Campus, Ashburn). *aDN-Gal4* (*Hampel et al., 2015*)(*R76F12-AD; R18C11-DBD*), was provided by J. Simpson (University of California, Santa Barbara). Wild-type, *PR* animals were provided by M. Dickinson (California Institute of Technology, Pasadena).

### Optogenetic stimulation experiments

Experiments were performed in the late morning or early afternoon Zeitgeber time (Z.T.), inside a dark imaging chamber. An adult female animal 2–3 days-post-eclosion (dpe), was mounted onto a custom stage (*Chen et al., 2018*) and allowed to acclimate for 5 min on an air-supported spherical treadmill (*Chen et al., 2018*). Optogenetic stimulation was performed using a 617 nm LED (Thorlabs, Newton, NJ) pointed at the dorsal thorax through a hole in the stage, and focused with a lens (LA1951, 01" f = 25.4 mm, Thorlabs, Newton, NJ). Tethered flies were otherwise allowed to behave spontaneously. Data were acquired in 9 s epochs: 2 s baseline, 5 s with optogenetic illumination, and 2 s without stimulation. Individual flies were recorded for five trials each, with one-minute intervals. Data were excluded from analysis if flies pushed their abdomens onto the spherical treadmill—interfering with limb movements—or if flies struggled during optogenetic stimulation, pushing their forelimbs onto the stage for prolonged periods of time.

## Unsupervised behavioral classification

To create unsupervised embeddings of behavioral data, we mostly followed the approach taken by *Todd et al. (2017)* and *Berman et al. (2014)*. We smoothed 3D pose traces using a 1€ filter. Then we converted them into angles to achieve scale and translational invariance (*Casiez et al., 2012*). Angles were calculated by taking the dot product from sets of three connected 3D positions. For

the antennae, we calculated the angle of the line defined by two antennal points with respect to the ground-plane. This way, we generated four angles per leg (two body-coxa, one coxa-femur, and one femur-tibia), two angles for the abdomen (top and bottom abdominal stripes), and a single angle for the antennae (head tilt with respect to the axis of gravity). In total, we obtained a set of 20 angles, extracted from 38 3D points.

We transformed angular time series using a Continous Wavelet Transform (CWT) to create a posture-dynamics space. We used the Morlet Wavelet as the mother wavelet, given its suitability to isolate periodic chirps of motion. We chose 25 wavelet scales to match dyadically spaced center frequencies between 5 Hz and 50 Hz. Then, we calculatd spectrograms for each postural time-series by taking the magnitudes of the wavelet coefficients. This yields a $20 \times 25 = 500$-dimensional time-series, which was then normalized over all frequency channels to unit length, at each time instance. Then, we could treat each feature vector from each time instance as a distribution over all frequency channels.

Later, from the posture-dynamics space, we computed a two-dimensional representation of behavior by using the non-linear embedding algorithm, t-SNE (*Maaten, 2008*). t-SNE embedded our high-dimensional posture-dynamics space onto a 2D plane, while preserving the high-dimensional local structure, while sacrificing larger scale accuracy. We used the Kullback–Leibler (KL) divergence as the distance function in our t-SNE algorithm. KL assesses the difference between the shapes of two distributions, justifying the normalization step in the preceding step. By analyzing a multitude of plots generated with different perplexity values, we empirically found a perplexity value of 35 to best suit the features of our posture-dynamics space.

From this generated discrete space, we created a continuous 2D distribution, that we could then segment into behavioral clusters. We started by normalizing the 2D t-SNE projected space into a $1000 \times 1000$ matrix. Then, we applied a 2D Gaussian convolution with a kernel of size $\sigma = 10$ px. Finally, we segmented this space by inverting it and applying a Watershed algorithm that separated adjacent basins, yielding a behavioral map.

## Acknowledgements

We thank Celine Magrini and Fanny Magaud for image annotation assistance, Raphael Laporte and Victor Lobato Ríos for helping to develop camera acquisition software.

## Additional information

### Funding

| Funder | Grant reference number | Author |
|---|---|---|
| Schweizerischer Nationalfonds zur Förderung der Wissenschaftlichen Forschung | 175667 | Daniel Morales Pavan Ramdya |
| Schweizerischer Nationalfonds zur Förderung der Wissenschaftlichen Forschung | 181239 | Daniel Morales Pavan Ramdya |
| EPFL | iPhD | Semih Günel |
| Microsoft Research | JRC Project | Helge Rhodin |
| Swiss Government Excellence Postdoctoral Scholarship | 2018.0483 | Daniel Morales |

The funders had no role in study design, data collection and interpretation, or the decision to submit the work for publication.

### Author contributions

Semih Günel, Conceptualization, Data curation, Software, Formal analysis, Validation, Investigation, Visualization, Methodology, Writing—original draft, Writing—review and editing; Helge Rhodin, Conceptualization, Software, Formal analysis, Supervision, Methodology, Writing—original draft, Project administration, Writing—review and editing; Daniel Morales, Data curation, Investigation,

Writing—review and editing; João Campagnolo, Data curation, Software, Writing—review and editing; Pavan Ramdya, Conceptualization, Resources, Supervision, Funding acquisition, Methodology, Writing—original draft, Project administration, Writing—review and editing; Pascal Fua, Conceptualization, Resources, Supervision, Funding acquisition, Methodology, Project administration, Writing—review and editing

### Author ORCIDs
Helge Rhodin http://orcid.org/0000-0003-2692-0801
Daniel Morales http://orcid.org/0000-0002-7469-0898
Pavan Ramdya https://orcid.org/0000-0001-5425-4610

### Decision letter and Author response
Decision letter https://doi.org/10.7554/eLife.48571.032
Author response https://doi.org/10.7554/eLife.48571.033

## Additional files

### Supplementary files
• Transparent reporting form DOI: https://doi.org/10.7554/eLife.48571.022

### Data availability
All data generated and analyzed during this study are included in the DeepFly3D GitHub site: https://github.com/NeLy-EPFL/DeepFly3D (copy archived at https://github.com/elifesciences-publications/DeepFly3D) and in the Harvard Dataverse.

The following datasets were generated:

| Author(s) | Year | Dataset title | Dataset URL | Database and Identifier |
|---|---|---|---|---|
| Gunel S, Rhodin H, Morales D, Campagnolo J, Ramdya P, Fua P | 2019 | aDN-GAL4 Control | https://doi.org/10.7910/DVN/PKKXOE | Harvard Dataverse, 10.7910/DVN/PKKXOE |
| Gunel S, Rhodin H, Morales D, Campagnolo J, Ramdya P, Fua P | 2019 | MDN-GAL4 Control | https://doi.org/10.7910/DVN/HOLXOR | Harvard Dataverse, 10.7910/DVN/HOLXOR |
| Gunel S, Rhodin H, Morales D, Campagnolo J, Ramdya P, Fua P | 2019 | aDN-GAL4 UAS-CsChrimson | https://doi.org/10.7910/DVN/S4L4KX | Harvard Dataverse, 10.7910/DVN/S4L4KX |
| Gunel S, Rhodin H, Morales D, Campagnolo J, Ramdya P, Fua P | 2019 | MDN-GAL4 UAS-CsChrimson | https://doi.org/10.7910/DVN/8SUC9U | Harvard Dataverse, 10.7910/DVN/8SUC9U |

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
