## [Decision Letter]

Thank you for submitting your article "DeepFly3D, a deep learning-based approach for 3D limb and appendage tracking in tethered, adult *Drosophila*" for consideration by *eLife*. Your article has been reviewed by two peer reviewers, and the evaluation has been overseen by a Reviewing Editor and Ronald Calabrese as the Senior Editor. The following individuals involved in review of your submission have agreed to reveal their identity: Josh W Shaevitz (Reviewer #2).

The reviewers have discussed the reviews with one another and the Reviewing Editor has drafted this decision to help you prepare a revised submission.

Both reviewers agreed that this tools and resources article can potentially be of use to the field. By automating pose tracking a large amount of data can be extracted from experiments, enabling new scientific questions to be addressed.

However, a number of key issues with the current study were identified and outlined in the reviewer comments below. In summary, the current manuscript:

1) Lacks a comprehensive discussion of how experimental choices, e.g. camera number, affect tracking error

2) Reports a very large tracking error using a much larger training set than current state-of-the-art 2D tracking algorithms. This needs to be accounted for.

3) Lacks some key references in the field and comparison with other approaches.

Reviewer #1:

Günel et al. report the development of a new toolkit to perform 3D pose estimation on tethered flies. While the toolkit is certainly bound to find utility, there are not major computational advances or insights within the work. While deep learning approaches for animal pose estimation is a "young field" in neuroscience/ethology (i.e. (Pereira et al., 2019, Mathis et al., 2018, Nath et al., 2019), it would be important for the authors to clarify their contribution in light of this. Namely, the authors should clearly discuss the major points that potential users care about: (1) accuracy of the tool, (2) amount of data required to train networks, if required, (3) speed, and (4) usability.

While DeepFly3D provides very nice user interfaces for the toolkit, it does suffer in the other three main categories.

1) Accuracy. The authors report a 13.9-pixel error on 480 by 920 images, and consider the PCK to be 50 pixels, which is nearly as long as a leg segment – not the width! I would argue this is far too lenient of a threshold. Additionally, two points the authors need to address (1) The RMSE would be more meaningful with an estimate of the human labeling variability (i.e. what is the best the network could do?), and (2) why, given so much training data, is the performance not as accurate as other approaches (LEAP, DeepLabCut)?

i.e. in reference to:

– "… 480× 960 pixels. The test dataset included challenging frames and occasional motion blur to increase the difficulty of pose estimation. For training, we used a final training dataset of 37,000 frames, an overwhelming majority of which were first automatically corrected using pictorial structures. On test data, we achieved a Root Mean Square Error (RMSE) of 13.9 pixels"

– The error is quite a bit larger than other animal pose estimation approaches, i.e. on 480x640 on 3D freely moving flies DeepLabCut used ~560 frames with ~4-pixel error on 682 × 540 frames. LEAP: ~500 images to achieve ~3-pixel error on 192 × 192-pixel frames with flies. While I appreciate that a direct comparison is not fully straightforward, and the authors do not use "easy" points to compute the pixel error, the authors absolutely should discuss/address this point.

2) The amount of data. As this network was trained on ~37K images, it is likely expected that the end-user will use these authors pre-trained network or the human pre-trained network. Namely, creating a large dataset seems infeasible when other options exist that require much less data. Namely, DeepLabCut reports 50-200 for most applications, and LEAP requires only ~500-1,500, which is nearly a factor of 10 less images to match similar performance (I understand that 2K are hand-labeled, while the rest are not, but even that is a 4X increase over the best method, i.e. DeepLabCut). Thus, the amount of training data for the performance is very high, given current alternatives. These limitations should be discussed.

3) Speed. There is no report of the speed of the network, thus the authors should benchmark the performance on video analysis speed (only network training time is reported).

4) Since the authors needed a lot of labeled data, making this a most useful as a specialized network for tethered flies, i.e. much like OpenPose has specific networks for hands (which they could clarify). Therefore, if they provide the network weights they should show the network is useable for other labs data.

5) 3D pose in the literature. The authors do not discuss Nath et al., which is a 3D extension of DeepLabCut (bioRxiv Nov 2018, and Nature Protocols, 2019) and given it's a small field, the authors should not claim theirs is the first to do 3D pose. Beyond deep learning multiple 3D pose estimation papers exist in Neuroscience (ie. Mimica et al., 2018 – Efficient cortical coding of 3D posture in freely behaving rats).

Therefore language like this should be modified:

"However, algorithms for reliably estimating 3D pose in such small animals have not yet been developed.…"

"However, these measurements have been restricted to 2D pose in freely behaving animals"

"We demonstrate more accurate unsupervised behavioral embedding using 3D joint angles rather than commonly used 2D pose data. Thus, DeepFly3D enables the automated acquisition of behavioral measurements at an unprecedented level of resolution for a variety of biological application"

(also here "resolution" should be edited; it is not that DeepFly3D is more accurate than other alternatives.)

However, I do think these authors have a nice solution, so I do find it very fair that they state they have an alternative 3D approach, (and can highlight the advantages) solution to multi-camera estimation in body- or head- fixed animals. However, again, sparse bundle adjustment is not a new technique, so the authors should discuss this in light of the literature.

6) "Here we present DeepFly3D, a software that infers the 3D pose of tethered, adult *Drosophila*-or other animals-using multiple camera images" – they don't quantify performance for other animals, so this should be dropped from the Abstract.

7) "DeepLabCut provides a user-friendly interface to DeepCut, a state-of-the-art human pose estimation network" (DeepCut should be DeeperCut, which is different than DeepCut).

8) "Calibration without targets" is misleading. The authors do calibrate; they just use the rigid body of the fly that has set distances per limb segment. To note, this isn't always doable for other animals/videos, again another argument to drop the random "this can be used for anything" language…

Reviewer #2:

The manuscript by Gunel et al. describes a new tool for 3D pose tracking that automatically combines multiple 2D images from different viewing angles while also learning the camera model. There has been a flurry of recent papers that use deep neural networks to track pose in 2D images. This manuscript improves on these by generating 3D positions for each body part that is tracked using a combination of various algorithms that have been developed for human pose tracking. In addition, the authors provide a simple GUI that allows for part labeling and prediction correction. Overall the paper is well written and the most aspects of the system described and justified.

My only major issue is that the experimental setup is not discussed or motivated. In particular, I would like to see more information about the choices/requirements for the number of cameras and other considerations. Why do the authors use 7 cameras? I assume that with fewer cameras you get more error/occlusions/etc. Can you investigate these effects quantitatively? How should a researcher choose the number of cameras (recording from 7 cameras in synch is difficult)?

In addition, I would suggest that the authors show the distribution of 3D localization errors. Is this isotropic? Is it the same for all body parts (the Discussion subsection “Learning the parameters” suggests not)?

---

## [Author Response]

1) Lacks a comprehensive discussion of how experimental choices, e.g. camera number, affect tracking error2) Reports a very large tracking error using a much larger training set than current state-of-the-art 2D tracking algorithms. This needs to be accounted for.3) Lacks some key references in the field and comparison with other approaches.Reviewer #1:[…] While DeepFly3D provides very nice user interfaces for the toolkit, it does suffer in the other three main categories.1) Accuracy. The authors report a 13.9-pixel error on 480 by 920 images, and consider the PCK to be 50 pixels, which is nearly as long as a leg segment – not the width! I would argue this is far too lenient of a threshold.

We illustrate the relative sizes of the error threshold (50 pixels) and limb segment lengths in Author response image 1. We set the threshold to be approximately one third the length of femur (erroneously called the “femur tibia segment” in the initial submission), which is approximately 150 pixels. We apologize for any confusion causing the reviewer to interpret this as the width and have adjusted the text accordingly. We now state: “We set this threshold as 50 pixels, which is roughly one third of the 3D length of the femur.” If we reduce our threshold to 30 or 20 pixels, we still achieve 95% or 89% accuracy, respectively. (Author response image 1).

**Author response image 1. respfig1:** Interpreting and choosing the PCK Threshold. (**a**) An illustration of the length of the PCK (percentage of keypoints) error threshold (Scale bar, bottom right) for comparison with limb segment lengths. The 50 pixel threshold is approximately one-third the length of the prothoracic femur. (**b**) PCK accuracy as a function of mean absolute error (MAE) threshold.

Additionally, two points the authors need to address (1) The RMSE would be more meaningful with an estimate of the human labeling variability (i.e. what is the best the network could do?)

We agree with the reviewer, and have now added this important measurement by calculating the accuracy of human manual-annotations as a baseline for interpreting the quality of our network predictions. We randomly selected 210 images and had a new expert manually annotate them. We observed a RMSE of 12.4 pixels. Therefore, our Network Annotation / Manual Annotation ratio of 1.12 (13.9 pixels / 12.4 pixels) is similar to the ratio reported by Mathis et al., 2018: 1.07 (2.88 pixels / 2.69 pixels) (pg. 1282 the last paragraph, and pg. 1284 the first paragraph). However, this ratio is reported for a mouse dataset, and human variability for Drosophila is not reported. We now include this information in the revised manuscript: “Compared with a ground truth RMSE of 12.4 pixels – via manual annotation of 210 images by a new human expert – our Network Annotation / Manual Annotation ratio of 1.12 (13.9 pixels / 12.4 pixels) is similar to another state-of-the-art network (Mathis et al., 2018): 1.07 (2.88 pixels / 2.69 pixels).”

[…] and (2) why, given so much training data, is the performance not as accurate as other approaches (LEAP, DeepLabCut)?i.e. in reference to:– "… 480× 960 pixels. The test dataset included challenging frames and occasional motion blur to increase the difficulty of pose estimation. For training, we used a final training dataset of 37,000 frames, an overwhelming majority of which were first automatically corrected using pictorial structures. On test data, we achieved a Root Mean Square Error (RMSE) of 13.9 pixels"– The error is quite a bit larger than other animal pose estimation approaches, i.e. on 480x640 on 3D freely moving flies DeepLabCut used ~560 frames with ~4-pixel error on 682 × 540 frames. LEAP: ~500 images to achieve ~3-pixel error on 192 × 192-pixel frames with flies. While I appreciate that a direct comparison is not fully straightforward, and the authors do not use "easy" points to compute the pixel error, the authors absolutely should discuss/address this point.

We understand that a 13.9-pixel error may seem large when compared with other pose estimation approaches. However, as the reviewer appreciates, making a direct comparison between datasets is not fully straightforward and we believe this misconception can be explained by several differences between the various datasets:

Comparison with the LEAP dataset:

1) The ~3-pixel error reported for 192x192-pixel images in Pereira et al., 2019 (LEAP) corresponds to a 11.9-pixel≅3/2(480/192)2+(960/192)2 error for the 480x960 pixel images used in our study if we similarly rescale pose predictions and ground truth annotations. See the Appendix below for a more detailed calculation of the scaling coefficient.

2) The LEAP dataset is relatively highly preprocessed compared with our dataset: images are rotated, aligned, and thresholded to generate a featureless, static background. By contrast, our data are raw, without any further preprocessing, and therefore harder for performing pose estimation. Our multiview dataset does not lend itself as readily to preprocessing, due to the more dynamic background including a rotating spherical treadmill and legs moving on the opposite side of the body.

Comparison with the DeepLabCut dataset:

1) Although Mathis et al., 2018 perform pose estimation on larger images (682x540 pixels), flies occupy a much smaller region in each image, (~340x135 pixels when flies are parallel to the image plane and take up the largest image area). By contrast, in our dataset, flies occupy nearly the entire image (~840x320 pixels). Thus, scaling the reported error for DeepLabCut (4 pixels for 340x135 pixel images) to 840x320 pixel images results in an error of 10.1 pixels ≅4.17/2(320/135)2+(840/340)2. This value is much closer to the 13.9 pixel error that we report for DeepFly3D.

2) Mathis et al., 2018 principally illustrates the tracking of landmarks on the fly’s abdomen and head (with only one landmark on the leg). These larger body parts experience smaller deformations and perform less rapid, complex movements than the legs.

These calculations are approximate but we intend for them to provide context for the error values we report in our manuscript. In short, we believe the seemingly large tracking error of DeepFly3D is (i) due to the scale of the images (rescaled pixel-wise error will grow with increasing image size) and (ii) due to the more visually complex nature of our dataset. It should be stressed that the 2D pose module of DeepFly3D – our Stacked Hourglass network – can be replaced with any other network. This includes the networks from Mathis et al., 2018, and Pereira et al., 2019, or any other pose estimation approach as long as the network outputs pixel-wise conditional probabilities.

The relative strengths of the Stacked Hourglass network and DeepLabCut have also been discussed in a recent preprint (Graving et al., 2019; Appendix 0 Figure 7). They show that, although DeepLabCut is more accurate than the Stacked Hourglass network (15% less error on a Drosophila dataset with the same number of annotations, ~2 pixel MAE versus ~2.3 pixel MAE using the same amount of annotation), the Stacked Hourglass network is much faster than DeepLabCut (4x on average). We believe this speed-accuracy trade-off is necessary to process 700 images produced per second in our system. Similarly, LEAP is significantly less accurate than both DeepLabCut and the Stacked Hourglass network on many different datasets (Graving et al., 2019; Appendix 0 Figure 7).

2) The amount of data. As this network was trained on ~37K images, it is likely expected that the end-user will use these authors pre-trained network or the human pre-trained network. Namely, creating a large dataset seems infeasible when other options exist that require much less data. Namely, DeepLabCut reports 50-200 for most applications, and LEAP requires only ~500-1,500, which is nearly a factor of 10 less images to match similar performance (I understand that 2K are hand-labeled, while the rest are not, but even that is a 4X increase over the best method, i.e. DeepLabCut). Thus, the amount of training data for the performance is very high, given current alternatives. These limitations should be discussed.

In principle, our Stacked Hourglass network also does not require a large amount of data to perform well. However, like all deep models, it gradually improves in performance and generalizes better with increasing amounts of data. Thus, we used a large training set to increase our network’s robustness and utility. Importantly, our DeepFly3D toolset is designed to simplify and automate annotation such that the large-dataset high-accuracy regimes can be reached relatively easily.

To illustrate these assertions and test the performance of the network in a low data regime, we trained a single 2 stacked network using ground-truth annotations data from 7 cameras. In Figure 2B, we compare the results with an asymptotic prediction error (i.e., the error observed when the network is trained using the full dataset of 40,000 annotated images). We also compare the results to human annotation variability (‘manual annotation error’). In this particular dataset, we observed an asymptotic MAE of 10.5 pixels and a human variability MAE of 9.2 pixels. Thus, the ratio between human variability remains close to 1.12. We observe that, with 800 annotations, our network achieves a similar accuracy to manual annotation error and the asymptotic prediction error. Further annotation only generates diminishing returns.

3) Speed. There is no report of the speed of the network, thus the authors should benchmark the performance on video analysis speed (only network training time is reported).

We agree with the reviewer that this may be an important piece of information for the end-user. Using the desktop configuration described in our original submission, we found that our network can run at 110 Frames-Per-Second (FPS) using the 8-stack variant of the Stacked Hourglass network, and can run at 420FPS when using the smaller 2-stack variant. Thanks to our effective initialization step, we observed that calibration takes only 3-4 s. We also measured the time required for error checking and error correction steps. Error checking can be done at 100 FPS and error correction can be done in 10 FPS. Since error correction is scarcely performed in response to a large reprojection error, it does not create a bottleneck in the overall speed of the pipeline.

We now include this information in the revised manuscript as follows: “Using this desktop configuration, our network can run at 100 Frames-Per-Second (FPS) using the 8-stack variant of the Hourglass network, and can run at 420 FPS using the smaller 2-stack version. Thanks to an effective initialization step, calibration takes 3-4 s. Error checking and error correction can be performed at 100 FPS and 10 FPS, respectively. Error correction is only performed in response to large reprojection errors, and does not create a bottleneck in the overall speed of the pipeline.”

4) Since the authors needed a lot of labeled data, making this a most useful as a specialized network for tethered flies, i.e. much like OpenPose has specific networks for hands (which they could clarify). Therefore, if they provide the network weights they should show the network is useable for other labs data.

As illustrated in our earlier response to point #2, our Stacked Hourglass network does not require 37K images to achieve an accuracy comparable with that reported in Mathis et al., 2018. We have publicly published our dataset (https://dataverse.harvard.edu/dataverse/DeepFly3D), but we do not have access to video data from a synchronized, multi-camera system and thus could not test our network on another laboratory’s data. We would also like to emphasize that DeepFly3D is specialized for tethered flies but that the computational tools behind our data analysis pipeline are general, as demonstrated by our adaptation of DeepFly3D to the human H3.6m dataset (Figure 11).

5) 3D pose in the literature. The authors do not discuss Nath et al., which is a 3D extension of DeepLabCut (bioRxiv Nov 2018, and Nature Protocols, 2019) and given it's a small field, the authors should not claim theirs is the first to do 3D pose. Beyond deep learning multiple 3D pose estimation papers exist in Neuroscience (ie. Mimica et al., 2018 – Efficient cortical coding of 3D posture in freely behaving rats).

We thank the reviewer for pointing out relevant literature. We now also cite Nath et al., 2019. However, we would like to point out that we do not claim that DeepFly3D is the first approach to generally perform 3D pose estimation. We recognize a rich human pose estimation literature and an increasing number of studies performing pose estimation on large laboratory animals (including rodents). These studies achieve 3D pose by triangulating multiple camera views using, for example a checkerboard pattern, and readily available software. To the best of our knowledge, this checkerboard-based registration approach is what is used by the OpenCV library that is referred to in Nath et al., 2019: “3D kinematics can be reconstructed from one network being trained on multiple views, and the user needs only the camera calibration files to reconstruct the data. This camera calibration and triangulation can be done in many programs, including the free package OpenCV”.

By contrast, by developing DeepFly3D, we tried to be careful to point out that we are performing 3D pose estimation on uniquely challenging fly-sized objects. To make this clear, we stated that “techniques used to translate human 2D pose to 3D pose cannot be easily transferred for the study of small animals like Drosophila” because “precisely registering multiple camera viewpoints using traditional approaches would require the fabrication of a prohibitively small checkerboard pattern”. Therefore, we still contend that DeepFly3D provides a novel contribution to 3D pose estimation because “the unique challenges associated with transforming these 2D measurements into reliable and precise 3D poses have not been addressed for small animals including the fly, Drosophila melanogaster” and “algorithms for reliably estimating 3D pose in such small animals have not yet been developed.” Nevertheless, as described below we modified the language in our revised manuscript to further clarify these points.

Therefore language like this should be modified:"However, algorithms for reliably estimating 3D pose in such small animals have not yet been developed.…"

We agree that our definition of “small” may have been unclear, and in light of the reviewer’s comments, we have edited this sentence. The manuscript now states: “However, algorithms for reliably estimating 3D pose in such small Drosophila-sized animals have not yet been developed.”

"However, these measurements have been restricted to 2D pose in freely behaving animals"

We agree with the reviewer that it was initially unclear that this sentence refers entirely to work on insects – for which this statement is correct. We have removed a misleading citation to human pose (Mori and Malik, 2006), added a more relevant Drosophila citation (Isakov et al., 2016), and edited two sentences to now read: “Although this approach works well on humans (Moeslund et al.), in smaller, Drosophila-sized animals markers likely hamper movements,…”, and “However, these measurements have been restricted to 2D pose in freely behaving flies.

"We demonstrate more accurate unsupervised behavioral embedding using 3D joint angles rather than commonly used 2D pose data. Thus, DeepFly3D enables the automated acquisition of behavioral measurements at an unprecedented level of resolution for a variety of biological application" (also here "resolution" should be edited; it is not that DeepFly3D is more accurate than other alternatives.)

We agree with the reviewer and have edited this sentence to improve clarity. Here our reference to ‘resolution’ refers to the additional information provided by the 3rd dimension – something that has not otherwise been achieved for Drosophila. We have now edited this sentence in the Abstract to say “Thus, DeepFly3D enables the automated acquisition of Drosophila behavioral measurements at an unprecedented level of detail for a variety of biological applications.”

However, I do think these authors have a nice solution, so I do find it very fair that they state they have an alternative 3D approach, (and can highlight the advantages) solution to multi-camera estimation in body- or head- fixed animals. However, again, sparse bundle adjustment is not a new technique, so the authors should discuss this in light of the literature.

We thank the reviewer for their positive appreciation of our approach. We have now added additional citations to the Discussion of our approach. In the same lines, we now state: “During the calibration process, we also employ sparse bundle adjustment methods, as previously used for human pose estimation.”

6) "Here we present DeepFly3D, a software that infers the 3D pose of tethered, adult Drosophila-or other animals-using multiple camera images" – they don't quantify performance for other animals, so this should be dropped from the Abstract.

We have removed this reference to “-or other animals-” from the Abstract.

7) "DeepLabCut provides a user-friendly interface to DeepCut, a state-of-the-art human pose estimation network" (DeepCut should be DeeperCut, which is different than DeepCut).

We thank the reviewer for spotting this error. We changed ‘DeepCut’ to ‘DeeperCut’ in the revised manuscript.

8) "Calibration without targets" is misleading. The authors do calibrate; they just use the rigid body of the fly that has set distances per limb segment. To note, this isn't always doable for other animals/videos, again another argument to drop the random "this can be used for anything" language…

We apologize for the confusion. To clarify, our manuscript mentions “Calibration without external targets” and uses the word ‘external’ explicitly to distinguish calibrating using the fly itself versus some external device fabricated separately for this purpose (e.g., a checkerboard pattern). To fix this possible source of confusion, we have now changed the text to read “Calibration without an external calibration pattern".

Notably, our calibration procedure does not depend on a fixed limb segment length or a rigid body, but rather on the assumption that multiple cameras can see the same 3D point. However, a constant limb segment constraint is used during the automatic correction step, after the calibration. If a non-rigid body is present, we suggest discarding the segment length energy and only using the reprojection error and probability maps during the optimization. This can be done by setting the lambda_bone parameter to 0 inside the Config.py file.

Reviewer #2:The manuscript by Gunel et al. describes a new tool for 3D pose tracking that automatically combines multiple 2D images from different viewing angles while also learning the camera model. There has been a flurry of recent papers that use deep neural networks to track pose in 2D images. This manuscript improves on these by generating 3D positions for each body part that is tracked using a combination of various algorithms that have been developed for human pose tracking. In addition, the authors provide a simple GUI that allows for part labeling and prediction correction. Overall the paper is well written and the most aspects of the system described and justified.

We thank the reviewer for their positive appreciation of our work.

My only major issue is that the experimental setup is not discussed or motivated. In particular, I would like to see more information about the choices/requirements for the number of cameras and other considerations. Why do the authors use 7 cameras? I assume that with fewer cameras you get more error/occlusions/etc. Can you investigate these effects quantitatively? How should a researcher choose the number of cameras (recording from 7 cameras in synch is difficult)?

We agree with the reviewer that this is important information for an end-user. Triangulation requires that any 3D point be seen by at least two cameras. Therefore, to see landmarks on both sides of the animal, one would need to use at least 4 cameras – 2 cameras on each side of the animal. In Author response image 2 we examine the accuracy of 3D pose estimation as a function of number of cameras used for calibration and/or prediction.

**Author response image 2. respfig2:** Triangulation error for each joint as a function of number of cameras used for calibration. Mean absolute error (MAE) as a function of number of cameras used for triangulation and calibration, as well as human annotation versus network prediction. Calibration using all 7 cameras and human annotation is the ground truth (red circles). This ground truth is compared with using N cameras for triangulation and performing calibration from N cameras using either human annotation of images (blue circles) or network predictions of images (yellow circles). A comparison across joints (individual panels) demonstrates that certain joints (e.g., tarsus tips) are more susceptible to increased errors with fewer cameras.

Our 3D ground truth data are based on human annotated 2D points calibrated using all 7 cameras, and triangulated again using human annotated 2D points. We observe that 3D error doubles as we reduce the number of cameras from 7 to 4. Thus, 3 cameras could be removed from the setup, but this would lead to an approximately 0.05 mm increase in mean error.

Specifically we observe that: (i) Calibration is not a significant source of error. Calibrating with N cameras (blue circles), or 7 cameras (red circles) is not very different, (ii) More cameras lead to improved triangulation. With fewer cameras, even having calibrated using all 7 cameras (red circles), results in larger error. This may be because the camera views are sufficiently different, having largely non-overlapping 2D-detection failure cases. Thus, the redundancy provided by having more cameras mitigates detection errors by finding a 3D pose that is consistent across at least two camera views. (iii) The remaining error (distance between blue and orange circles) can be explained by the 2D network detections having a larger 2D error (13.9 pixels vs. 12.4 pixels RMSE with human annotations).

On the other hand, as the reviewer points out, there are practical tradeoffs for increasing the number of cameras. Ultimately, one could potentially reduce the number of cameras from seven to two (one on each side of the animal) by training a neural network that “lifts” 2D pose into 3D from single camera images (Martinez et al., 2017).

We have now added this additional information in the revised manuscript as follows: “We assessed the optimal number of cameras for DeepFly3D and concluded that increasing the number of cameras increases accuracy by stabilizing triangulation. Specifically, we observed the following. (i) Calibration is not a significant source of error: calibrating with fewer than 7 cameras does not dramatically increase estimation error. (ii) Having more cameras improves triangulation. Reducing the number of cameras down to four, even having calibrated with 7 cameras, results in an increase of ~0.05 mm triangulation error. This may be because the camera views are sufficiently different, having largely non-overlapping 2D-detection failure cases. Thus, the redundancy provided by having more cameras mitigates detection errors by finding a 3D pose that is consistent across at least two camera views.”

In addition, I would suggest that the authors show the distribution of 3D localization errors. Is this isotropic? Is it the same for all body parts (the Discussion subsection “Learning the parameters” suggests not)?

As mentioned, the error is not isotropic. We now plot all of the errors for each joint in Author response image 3. We believe that the tarsus tips exhibit larger error than the other joints due to the increased prevalence of occlusions from the spherical treadmill, and larger positional variance. On the other hand, increasing error for the body-coxa joints can be attributed to the difficulty of human annotation and occlusions from some perspectives.

**Author response image 3. respfig3:** MAE errors for limb and antennal landmarks. MAE as a function of landmark showing the differential distribution of errors. Violin plots are overlaid with raw data points (white circles).

**Appendix to Reviewers**

1) For a single point, the error can be decomposed into the error in the x and y dimensions separately, e.g. e=ex2+ey2. Assuming the error in the x and y dimensions can be defined by the same distribution, E[ex2+ey2]=E[2ex2]=2ex^=2ey^, where ex^ and ey^ are the means of the error distribution. Scaling the image by sx and sy in the x and y dimensions, respectively, results in the errores=(sxex)2+(syey)2 and therefore E[es]=E[(sxex)2+(syey)2]=(sx)2+(sy)2ex^=(sx)2+(sy)2ey^. Then the final error for the scaled image can be approximated by E[es]=E[e](sx)2+(sy)2/2.

A much simpler argument to understand the effects of image scaling on reported error – assuming that images are scaled by one factor in both dimensions – can be summarized as follows. On the LEAP dataset, using the scale factor of the y dimension to rescale the error would result in a 15 pixel error (3 pixels * 960/192). Similarly, using the scale factor of the x dimension would result in a 7.2 pixel error (3 pixels * 480/192). Our reported error, 13.9 pixels, is between these two values. In the same manner, on the DeepLabCut dataset, using the scale factor of the x dimension to rescale the error would result in a 9.9 pixel error (4.17 pixels * 320/135). Using the scale factor of the y dimension would result in a 10.3 pixel error (4 pixels * 840/340). These results are also similar to our reported error of 13.9 pixels.